# Giant Rashba splitting in PtTe/PtTe$_2$ heterostructure

**Runfa Feng** [1], **Yang Zhang** [1], **Jiaheng Li**[1], **Qian Li**[1], **Changhua Bao** [1], **Hongyun Zhang** [1,2], **Wanying Chen**[1], **Xiao Tang**[1], **Ken Yaegashi**[2], **Katsuaki Sugawara** [2,3], **Takafumi Sato** [2,3], **Wenhui Duan** [1,4,5], **Pu Yu** [1,4] ✉ & **Shuyun Zhou** [1,4] ✉

Achieving a large spin splitting is highly desirable for spintronic devices, which often requires breaking of the inversion symmetry. However, many atomically thin films are centrosymmetric, making them unsuitable for spintronic applications. Here, we report a strategy to achieve inversion symmetry breaking from a centrosymmetric transition metal dichalcogenide (TMDC) bilayer PtTe$_2$, leading to a giant Rashba spin splitting. Specifically, the thermal annealing turns one layer of PtTe$_2$ sample into a transition metal monochalcogenide (TMMC) PtTe through Te extraction, thus forming PtTe/PtTe$_2$ heterostructure with inversion symmetry breaking. In this naturally-formed PtTe/PtTe$_2$ heterostructure, we observe a giant Rashba spin splitting with Rashba coefficient of $\alpha_R = 1.8$ eV · Å, as revealed by spin- and angle-resolved photoemission spectroscopy measurements. Our work demonstrates a convenient and effective pathway for achieving pronounced Rashba splitting in centrosymmetric TMDC thin films by creating TMMC/TMDC heterostructure, thereby extending their potential applications to spintronics.

Symmetry breaking plays an important role in the microscopic physics of solid-state materials, which determines their macroscopic electrical, optical, and magnetic properties[1]. In particular, the inversion symmetry breaking can induce an internal electric field in materials, which leads to the splitting of spin-up and spin-down electronic states for systems with notable spin-orbit coupling (SOC)[2-5]. Such effect, namely Rashba effect, can lead to a conversion between spin and charge currents[4,6-9], which therefore lays a promising foundation for spintronic applications. For practical spintronic devices, achieving a large spin splitting in atomically thin films is highly desirable for obtaining a high spin-to-charge conversion efficiency[8,10,11]. While a giant Rashba effect has been found in a few three-dimensional (3D) bulk materials such as BiTeI[12] and GeTe[13,14], the Rashba effect decreases quickly in the two-dimensional (2D) limit. For instance, the Rashba effect is reduced to half from bulk to

monolayer in BiTeI[12,15], while for GeTe, it is reduced to nearly zero in the monolayer film[14,16,17]. Although of scientifically and technically importance, it is still challenging to introduce a large Rashba effect in atomically thin films.

Transition metal dichalcogenides (TMDCs) have demonstrated advantages in obtaining atomically thin films or flakes, as well as offering rich possibilities for constructing a wide variety of van der Waals heterostructures with exotic electronic states[18]. While inversion symmetry breaking is observed in a few TMDC monolayer films (e.g., MoS$_2$ and NbSe$_2$), many other monolayer 2D TMDCs, such as PtTe$_2$ and PtSe$_2$, are centrosymmetric. In these latter materials, the local (site) inversion symmetry breaking leads to an interesting local Rashba effect with a spin-layer locking[19,20], while the global band structure is spin degenerate. To achieve a pronounced (global) Rashba splitting, constructing various van der Waals

[1]State Key Laboratory of Low-Dimensional Quantum Physics and Department of Physics, Tsinghua University, Beijing, PR China. [2]Advanced Institute for Materials Research (WPI-AIMR), Tohoku University, Sendai, Japan. [3]Department of Physics, Graduate School of Science, Tohoku University, Sendai, Japan. [4]Frontier Science Center for Quantum Information, Beijing, PR China. [5]Institute for Advanced Study, Tsinghua University, Beijing, PR China. ✉e-mail: yupu@mail.tsinghua.edu.cn; syzhou@mail.tsinghua.edu.cn

heterostructures to break the inversion symmetry by leveraging the 2D nature of these materials has been proposed in heterostructures with dissimilar TMDCs[21–25]. However, constructing such complicated heterostructures remains a challenge to achieve for practical applications. In this work, we provide a pathway to tailor the inversion symmetry by taking the advantage of another characteristic feature of TMDCs, namely the volatile nature of the anions (e.g., Te, Se), in which the high temperature annealing induces anionic extraction form transition metal monochalcogenide (TMMC) at the surface, thereby forming atomically designed TMMC/TMDC heterostructure with tailored crystalline symmetry and Rashba spin splitting. Here, we use bilayer (2 ML) $PtTe_2$ film as an example to demonstrate this concept, in which a high-quality $PtTe/PtTe_2$ heterostructure is achieved by annealing the 2 ML $PtTe_2$ film in ultra-high vacuum (UHV). Such atomic stacking between PtTe and $PtTe_2$ naturally breaks the inversion symmetry, as confirmed by the second harmonic generation (SHG) measurements. Angle-resolved photoemission spectroscopy (ARPES) and spin-resolved ARPES (Spin-ARPES) measurements reveal a pronounced spin splitting with Rashba coefficient of 1.8 eV · Å, which surpasses that in TMDCs and other heterostructure, e.g., 1.0 eV · Å in $Bi_2Se_3/NbSe_2$ heterostructure[26]. Our work provides an important and convenient pathway for achieving a large Rashba splitting in atomically-thin films by constructing naturally-stacked van der Waals heterostructures.

## Results and discussion

### Conversion from $PtTe_2$ film to $PtTe/PtTe_2$ heterostructure

Bulk platinum dichalcogenides in the 1$T$ structure are type-II topological semimetals with highly-titled Dirac cones[27]. Atomically thin platinum dichalcogenide films can be grown by direct selenization (or tellurization) of platinum[28] or molecular beam epitaxy (MBE) growth[29–31]. The local dipole moments in such centrosymmetric films lead to local Rashba effect with spin-layer locking, where spin-up and spin-down electrons are locked into separated sublayers but degenerate in energy globally[20,30], as schematically illustrated in Fig. 1a, b. PtTe has the same in-plane atomic structure as $PtTe_2$, making it convenient to form a perfect $PtTe/PtTe_2$ heterostructure (Fig. 1c, see more in Supplementary Fig. 1) with a strong interfacial coupling. It is interesting to note that the inversion symmetry is broken in the $PtTe/PtTe_2$ heterostructure with the formation of an out-of-plane dipole. This together with the strong SOC of Pt could potentially lead to a Rashba splitting in $PtTe/PtTe_2$ heterostructure, as schematically illustrated in Fig. 1d.

1$T$-$PtTe_2$ film was grown on bilayer graphene (BLG) terminated 6H-SiC(0001) substrate by MBE method (see Supplementary Figs. 2, 3)[30]. Previous studies reveal that post-growth annealing in Te deficient atmosphere (e.g., UHV), $PtTe_2$ films can be converted into PtTe through Te extraction[32–34]. Here, we show that through carefully controlled annealing condition (duration and temperature), it is possible to achieve partial conversion from $PtTe_2$ into PtTe, thereby forming $PtTe/PtTe_2$ heterostructure (Fig. 2a). Since PtTe consists of two Pt layers while $PtTe_2$ consists of only one layer, the film coverage would decrease as a result of Pt atom conservation (see Supplementary Fig. 2d, e). We note that such control is highly repeatable, as demonstrated in a series of samples with different thickness (see Supplementary Fig. 4). To confirm the formation of new structure, we carried out Raman spectra measurements (see Supplementary Fig. 5 for more details), since different structures can lead to distinct vibrational modes. Figure 2b demonstrates that new peaks (indicated by red arrow in Fig. 2b) emerge in the Raman spectra of the annealing sample (red curve in Fig. 2b) compared to pristine $PtTe_2$ films (golden curve), which can be assigned to PtTe layers (green curve). With this result, it is clear that the $PtTe_2$ film is partially converted into PtTe layer, thereby forming $PtTe/PtTe_2$ heterostructure.

### Evidence of inversion symmetry breaking from SHG measurement

The inversion symmetry breaking in the $PtTe/PtTe_2$ heterostructure is confirmed by SHG measurement, a technique highly sensitive to the crystal symmetry[35]. Figure 2c shows the SHG signal as a function of the sample azimuthal angle, with the polarization of the SHG signal parallel (red) and perpendicular (purple) to that of the incident laser. A SHG signal with six-fold rotational symmetry is clearly observed, confirming the breaking of inversion symmetry in $PtTe/PtTe_2$ heterostructure. This is in sharp contrast to the bilayer $PtTe_2$ before annealing, where the centrosymmetric structure leads to zero nonlinear susceptibility $\chi^{(2)}$ with negligible SHG signal (see Supplementary Fig. 6). Figure 2d, e further shows the evolution of the SHG signal as a function of the film thickness using the parallel polarization detection geometry. The SHG intensity decreases abruptly when the thickness of the $PtTe/PtTe_2$ heterostructure increases (Fig. 2e), which should be attributed to partially recovered inversion symmetry in thicker sample (see Supplementary Fig. 7). Scanning transmission electron microscopy (STEM) measurement (Fig. 2f) provides clear evidence of transition from $PtTe_2$ to PtTe, where a sandwiched Te-Pt-Te ($PtTe_2$) transforms to quadruple layer Te-Pt-Pt-Te (PtTe). However, it is worth mentioning that stacking faults can be formed in such structure (see Supplementary Fig. 8), which could also lead to suppressed SHG signal due to the compensating global polarity. Nevertheless, the observation of strong SHG in the thinnest $PtTe/PtTe_2$ heterostructure together with STEM results confirm that the inversion symmetry is broken, providing a prerequisite condition to host pronounced Rashba effect.

### Rashba band splitting and spin texture

We expect that the modification in the crystal structure could lead to a drastic change in the electronic structure, which can be directly probed through ARPES techniques. Comparison of dispersion images measured along the high-symmetry directions M-$\Gamma$-K for bilayer $PtTe_2$ (Fig. 3a, b) and obtained $PtTe/PtTe_2$ heterostructure from annealing (Fig. 3c, d) shows several distinct features. For bilayer $PtTe_2$, the Fermi surface shows a large hole pocket centered at $\Gamma$ point, small electron pockets near the K point and midpoint between $\Gamma$ and M (Fig. 3b). While, in $PtTe/PtTe_2$ sample, we observe the emergence of additional bands near the Fermi energy, which should be attributed to the formed PtTe layer (Fig. 3e, f). Furthermore, the Fermi surface map of $PtTe/PtTe_2$ heterostructure shows more pockets, with one circular hole pocket centered at the $\Gamma$ point, and a few larger pockets with clear warping (Fig. 3d). At high binding energy, e.g., 1 eV, the nearly parabolic band in Fig. 3a splits into two bands in Fig. 3c (pointed by box). These two bands are degenerate at the $\Gamma$ point and split when moving away from $\Gamma$, resembling Rashba-splitting bands as schematically illustrated in Fig. 1d, providing direct evidence for the inversion symmetry breaking induced Rashba effect in $PtTe/PtTe_2$ heterostructure.

Spin-ARPES measurements have been performed to verify the spin polarization of these splitting bands as shown in Fig. 4a, with the experimental configuration shown in Fig. 4b. The spin-resolved energy distribution curves (EDCs) shown in Fig. 4c, d represent measurement at two opposite momentum positions marked by black broken lines c, d in Fig. 4a. Clear spin intensity contrast shows that electrons are spin-polarized along the $y$ direction, and the two splitting bands have opposite spin polarizations. In contrast to the large spin polarization along the $y$ (in-plane) direction, the spin polarization is negligible along the $z$ (out-of-plane) direction, as shown in the lower panel of Fig. 4e. Figure 4h further shows two-dimensional spin contrast image measured as a function of both energy and momentum, which is obtained by taking the difference between the spin-up and spin-down intensities shown in Fig. 4f at

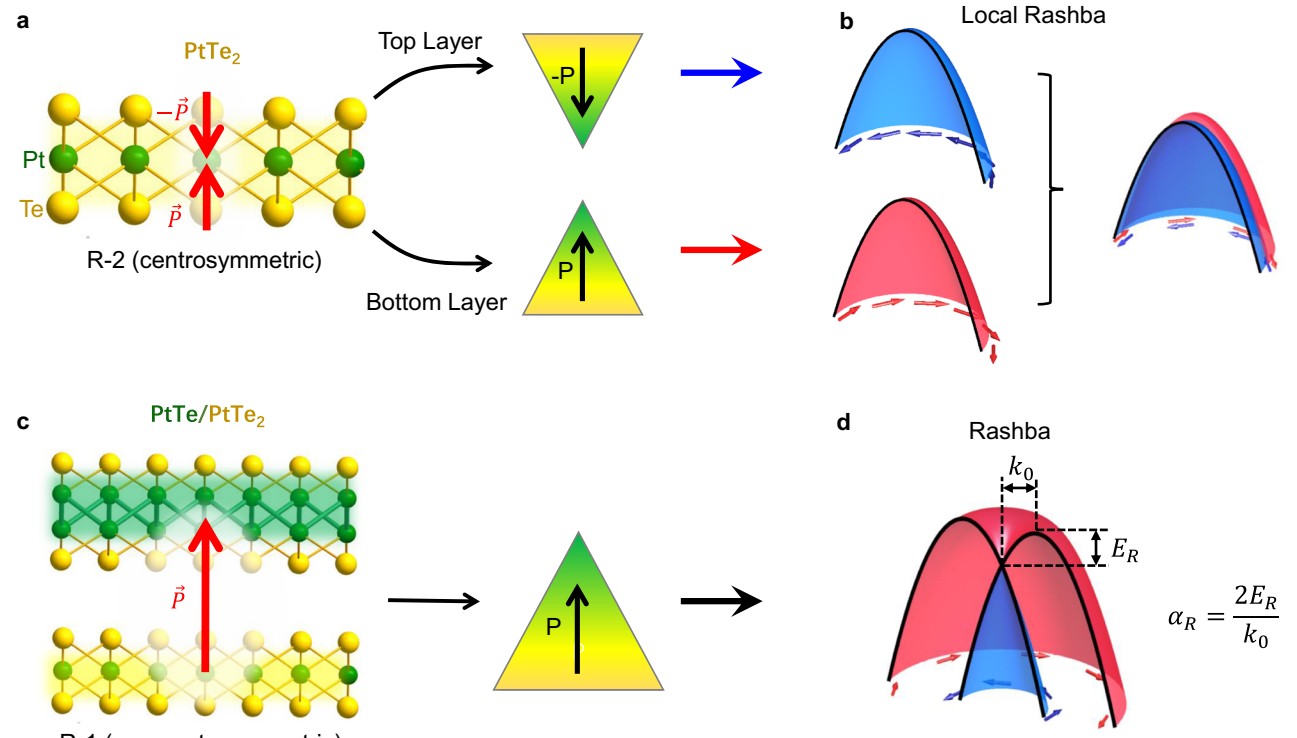

**Fig. 1 | Schematic illustration for local Rashba effect in a centrosymmetric PtTe$_2$ film and Rashba effect in a PtTe/PtTe$_2$ heterostructure. a** Crystal structure of monolayer PtTe$_2$, and schematic illustration for the local dipole moment **P**. **b** Local dipole moments with opposite directions induce helical spin texture with opposite helicities locked to the top and bottom Te layers respectively. The opposite spins are degenerate in energy. **c** Crystal structure of PtTe/PtTe$_2$ heterostructure, and schematic illustration for the net dipole moment. **d** The net dipole moment leads to Rashba-splitting bands. The Rashba effect can be quantified by Rashba coefficient $\alpha_R = 2E_R/k_0$, which is defined by the ratio between the energy splitting $E_R$ and the momentum offset $k_0$.

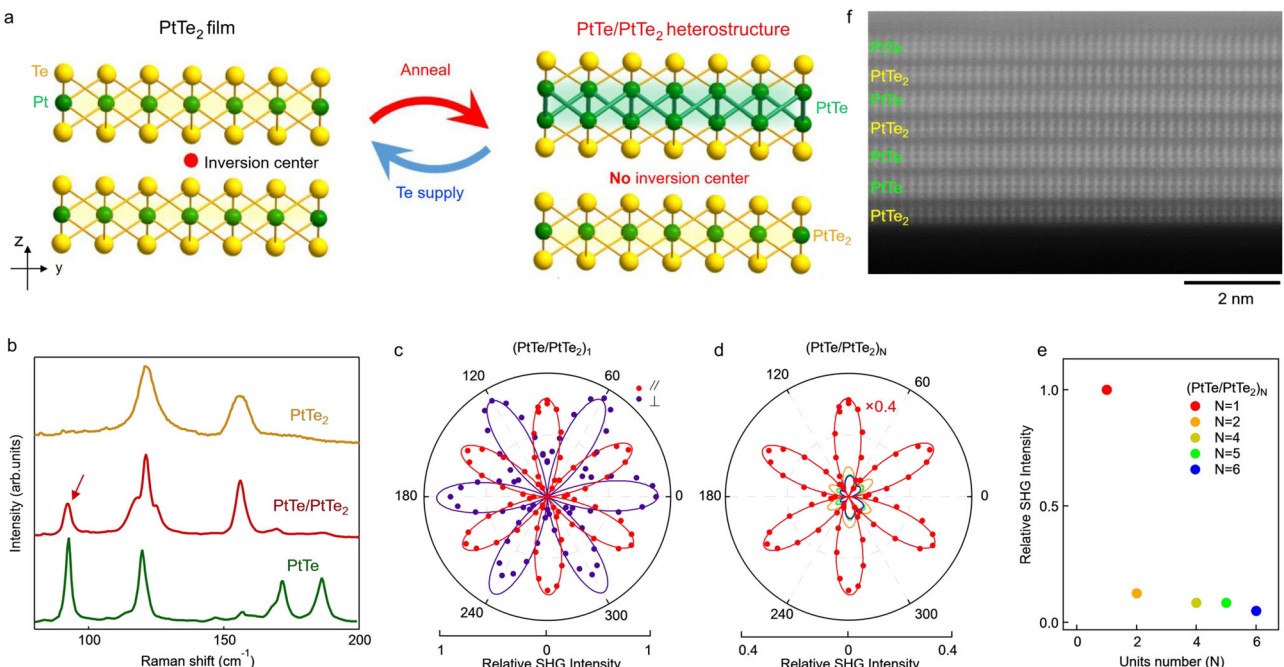

**Fig. 2 | Conversion from PtTe$_2$ to PtTe/PtTe$_2$ heterostructure. a** Crystal structure of PtTe$_2$ and PtTe/PtTe$_2$ (side view). **b** Raman spectra of PtTe$_2$, PtTe/PtTe$_2$ and PtTe films. **c** Rotational anisotropy SHG pattern of PtTe/PtTe$_2$ heterostructure consisting of 1 ML PtTe and 1 ML PtTe$_2$, with the polarization of SHG signal parallel (red) or perpendicular (purple) to the incident laser. **d** SHG signals measured in multi-layer PtTe/PtTe$_2$ heterostructures, which were obtained by annealing 2$N$ layers of PtTe$_2$ ($N$ is an integer). **e** Extracted SHG intensity measured on samples obtained by annealing 2$N$ layers of PtTe$_2$. The signal is normalized by the intensity of PtTe/PtTe$_2$ heterostructure. The error bar is smaller than the symbol size. **f** STEM image of converted multilayer PtTe/PtTe$_2$ heterostructure on bilayer graphene/SiC substrates.

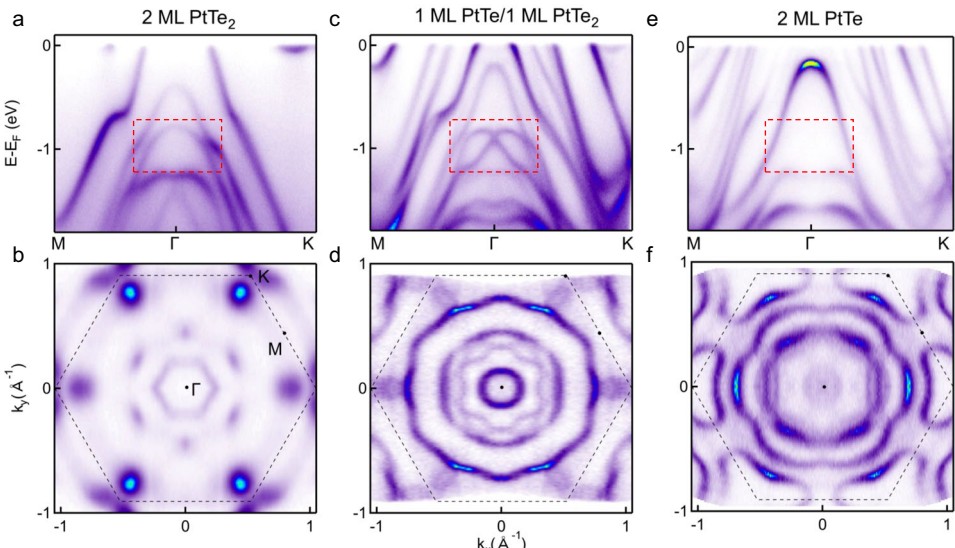

**Fig. 3 | Comparison of the electronic structure of bilayer PtTe₂ film, PtTe/PtTe₂ heterostructure and PtTe film. a** ARPES dispersion image of bilayer PtTe₂ measured along the M-Γ-K direction. The red box guides for the eyes to show the

dramatic modification of the band structure. **b** Constant energy maps at Fermi energies of bilayer PtTe₂. **c, d** The same as (**a, b**) but for PtTe/PtTe₂ heterostructure. **e, f** The same as (**a, b**) but for bilayer PtTe.

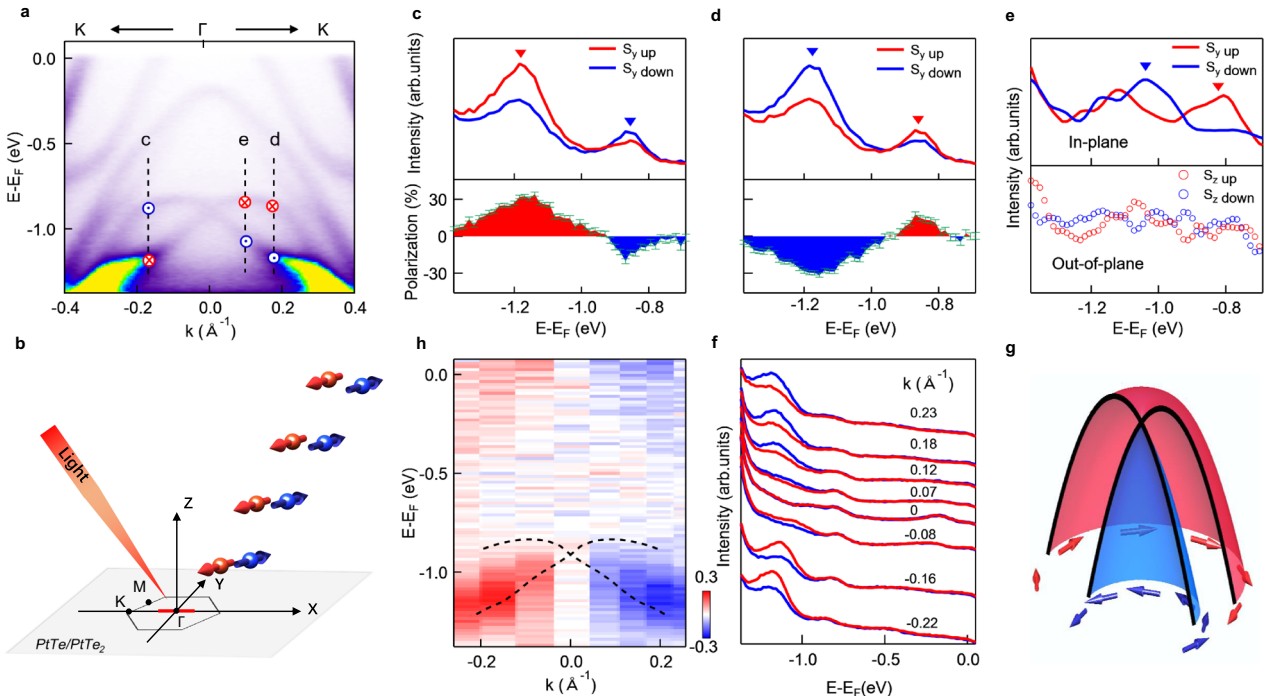

**Fig. 4 | Spin polarization of the Rashba-splitting bands. a** ARPES dispersion image measured on PtTe/PtTe₂ along the K-Γ-K direction ($h\nu = 60$ eV). Black broken lines mark the position for energy distribution curves (EDCs) shown in (**c–e**). The red crosses and blue dots stand for the spin directions. **b** Schematics of spin-ARPES measurement, where the red line shows the measurement direction. **c, d** Spin-resolved EDCs ($h\nu = 50$ eV). Red (blue) curves denote spin-up (spin-down)

components. Lower panel: corresponding extracted spin polarization, where the error bars are obtained from statistics of the measurements. **e** Spin-resolved EDCs to reveal the in-plane and out-of-plane components using Xe lamp ($h\nu = 8.4$ eV). **h** The spin-polarization image of Spin-ARPES ($h\nu = 60$ eV). **f** The corresponding EDCs of Spin-ARPES image in (**h**). **g** Schematic of the spin texture for the Rashba bands in the heterostructure.

different momentum. It further confirms that the spin-polarization has opposite directions across the Γ point. We also note that the spin contrast in the upper Rashba-splitting band is also confirmed with photon energy of 90 eV, which resolves the upper band better with much higher photoemission intensity due to the dipole matrix

element effect[36] (see Supplementary Fig. 9). We note that the matrix elements of the upper and lower bands are different because they have different contributions from PtTe and PtTe₂ layers. The above results confirm the Rashba splitting, where the inner and outer contours exhibit opposite helical spin textures (Fig. 4g).

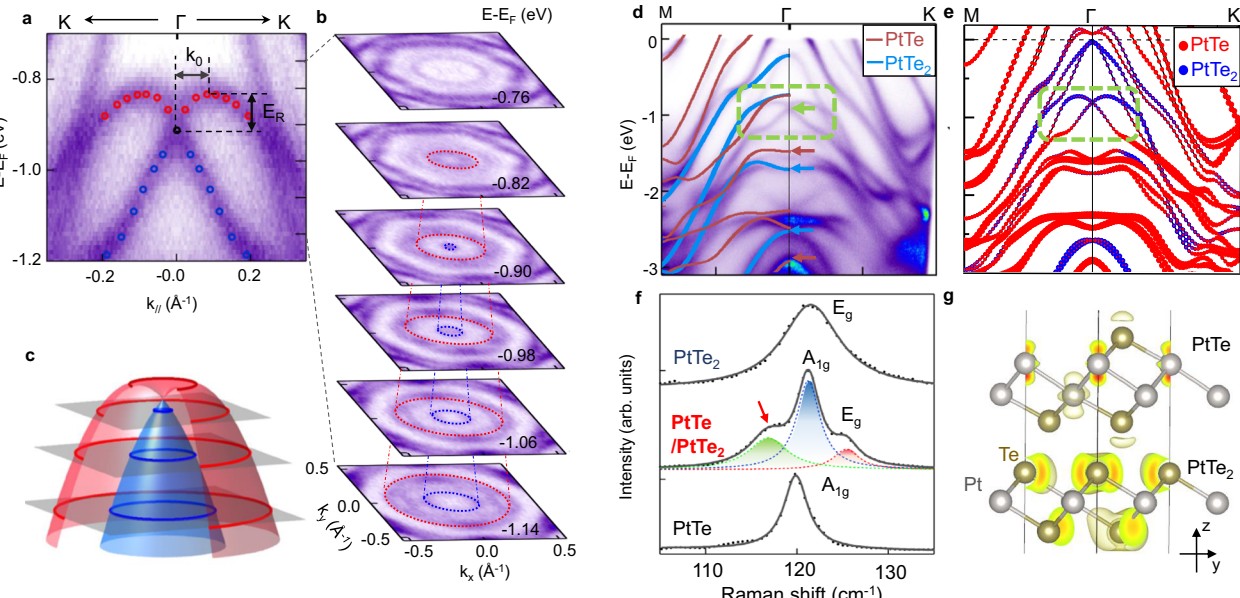

**Fig. 5 | Giant Rashba splitting in PtTe/PtTe$_2$ heterostructure and its origin.**
**a** Zoom-in dispersion image of Rashba-splitting bands, with energy splitting $E_R$ and the momentum offset $k_O$ labeled. **b** Constant energy contours to show Rashba-splitting bands. **c** Schematic summary for the Rashba-splitting bands.
**d** Comparison between experimental dispersion and calculated dispersion for monolayer PtTe (brown curve) and PtTe$_2$ (blue curve). Brown and blue arrows indicate electronic bands from PtTe and PtTe$_2$, respectively. **e** Calculated dispersion for PtTe/PtTe$_2$ heterostructure. Different colors and circle sizes distinguish contributions from PtTe and PtTe$_2$. **f** Raman spectra of PtTe$_2$, PtTe/PtTe$_2$ and PtTe, which is fitted by Lorentzian functions. **g** The real-space distributions of electric wavefunctions of the Rashba state at $\Gamma$ point.

**Table 1 | Representative two-dimensional Rashba materials and parameters characterizing their splitting strength: Rashba energy ($E_R$), momentum offset ($k_O$) and Rashba parameter ($\alpha_R$)**

| Materials | $E_R$(meV) | $k_O$(Å$^{-1}$) | $\alpha_R$(eV · Å) | Ref. |
|---|---|---|---|---|
| Metal surface states | | | | |
| Au(111) | 2.1 | 0.012 | 0.33 | Ref. 47 |
| Bi(111) | 14 | 0.05 | 0.55 | Ref. 48 |
| Ir(111) | | | 1.3 | Ref. 49 |
| 2D electron gas | | | | |
| InGaAs/InAlAs | <1 | 0.028 | 0.07 | Ref. 7 |
| KTaO$_3$/Al | | | 0.32 | Ref. 50 |
| Rb/Bi$_2$Se$_3$ | 52 | 0.08 | 1.3 | Ref. 51 |
| Heterostructure | | | | |
| Bi$_2$Se$_3$/NbSe$_2$ | 19 | 0.038 | 1.0 | Ref. 26 |
| **PtTe/PtTe$_2$** | **81** | **0.091** | **1.8** | **This work** |

### Rashba parameters

To quantify the strength of the Rashba effect in PtTe/PtTe$_2$, the Rashba coefficient is extracted. The zoom-in electronic structure in Fig. 5a clearly reveals the Rashba splitting. The corresponding energy contours in Fig. 5b further show two conical pockets with expanding size when moving downward in energy, confirming the Rashba splitting as schematically illustrated in Fig. 5c. The Rashba coefficient is extracted as $\alpha_R = 2E_R/k_O$, where $E_R$ and $k_O$ represent the energy and momentum scale of the splitting bands as labeled in Fig. 5a. By fitting the experimental dispersion, the extracted values are $E_R = 0.081 \pm 0.004$ eV and $k_O = 0.091 \pm 0.002$ Å$^{-1}$ (see Supplementary Fig. 10), which gives $\alpha_R = 1.8 \pm 0.2$ eV · Å. We note that the measured Rashba coefficient in PtTe/PtTe$_2$ is remarkable, as compared to representative two-dimensional Rashba materials including metal surface states, 2D electron gas, and heterostructures shown in Table 1.

The origin of the giant Rashba effect is further revealed by theoretical calculations. Figure 5d shows a comparison of ARPES dispersions with calculated band structures for monolayer PtTe (brown curves) and PtTe$_2$ (blue curves) films. It is clear that although the majority of bands resolved in the ARPES data can be associated with the bands from monolayer PtTe and monolayer PtTe$_2$ bands, several distinct features emerge, which should be attributed to the interlayer coupling/hybridization between PtTe and PtTe$_2$ layers (pointed by green arrow and green box). The calculated band structure for PtTe/PtTe$_2$ heterostructure (Fig. 5e) shows good agreement with the experimental dispersion image in Fig. 5d. Especially, the layer-resolved band structure shows clear Rashba bands within the green broken box, which are indeed contributed by both PtTe and PtTe$_2$ layer (Fig. 5e). Figure 5f shows a new structure emerging in the Raman spectra of PtTe/PtTe$_2$ heterostructure (see more detail in Supplementary Fig. 5), also indicating strong interlayer coupling between two layers. The charge redistribution in the heterostructure resulted from strong interlayer coupling which is responsible for the Rashba states, and the charge transfer between PtTe and PtTe$_2$ is verified by calculated real-space distributions of electric wavefunctions in the Rashba state (Fig. 5g). The doping dependent experiment further verifies this hypothesis. Upon K surface deposition, the charge injection from the top surface enhances the charge transfer and promotes the Rashba splitting, resulting in an increase of Rashba coefficient by 42% (see Supplementary Fig. 11). Therefore, both theoretical calculation and the Raman spectrum support the existence of Rashba splitting due to strong coupling between these two layers (Fig. 5g).

Here, we would like to further highlight the unique properties of the discovered giant Rashba effect in PtTe/PtTe$_2$ heterostructure. First, we find that the PtTe/PtTe$_2$ heterostructure can also be converted back to PtTe$_2$ with the formation of an extra PtTe$_2$ layer when annealing under Te flux (see Supplementary Fig. 12), providing a potentially reversible pathway to manipulate the inversion symmetry breaking with controllable Rashba splitting. Second, the PtTe/PtTe$_2$ heterostructure shows excellent stability with robust Rashba splitting under

exposure to the air for days (see Supplementary Fig. 13), which is critical for device application and indicates our ex-situ SHG measurement detects the intrinsic signal without surface oxidation as in PdTe$_2$ case[37]. Third, ARPES measurements on multilayer heterostructures, which are obtained by annealing thicker PtTe$_2$ films, shows a negligible Rashba splitting with thickness increasing (see Supplementary Fig. 4). This is consistent with the SHG measurements, where the strongest SHG signal is observed in the thinnest heterostructure consisting of monolayer PtTe and monolayer PtTe$_2$. Nevertheless, this result indicates the layer thickness form an exciting pathway to engineering the Rashba effect.

These results could have potential applications in spintronics devices. The giant Rashba states in PtTe/PtTe$_2$ could be possibly moved toward the Fermi level by Ir doping, as previous work has demonstrated the Fermi energy can be tuned in Ir$_x$Pt$_{1-x}$Te with Pt dopants while maintaining the same crystal structure and band feature[38]. Moreover, this strategy to induce a Rashba states may apply to a broad of materials that could be constructed into TMMC/TMDC heterostructure. For example, NiTe/NiTe$_2$[39] or CoTe/CoTe$_2$[40], in which they share similar symmetry and lattice constant, have the potential to obtain polar state towards Rashba effect along this pathway.

In this work, we introduce a strategy to obtain polar state in centrosymmetric PtTe$_2$ through thermal annealing induced PtTe/PtTe$_2$ heterostructure in which a giant Rashba effect is clearly resolved. The capability to control the inversion symmetry in TMDCs with strong SOC systems provides a convenient and efficient pathway for designing thin films with potential applications in next-generation nanoscale spintronic devices.

## Methods

### Thin film growth
PtTe$_2$ films were grown on bilayer-graphene (BLG)-terminated 6H-SiC(0001) substrates in a home-built MBE chamber with a base pressure of $5 \times 10^{-10}$ Torr. BLG/SiC substrates were prepared by flash annealing 6H-SiC(0001) to 1380 °C[41]. The smooth terrace of BLG was confirmed using scanning tunneling microscope (STM) (see Supplementary Fig. 2c). Pt from electron beam evaporator and Te from a Knudsen cell were deposited to substrates maintained at 300 °C. The flux ratio of Pt to Te was controlled to be ~1:50 to ensure a Te rich environment. The growth process was monitored by an in situ reflection high energy electron diffraction (RHEED) system and the growth rate was ~40 min per layer (extracted from RHEED oscillation in Supplementary Fig. 3). After growth, the sample was annealed at Te atmosphere for 20 min at 320 °C to achieve high-quality stoichiometric PtTe$_2$ samples. For fabrication of 1 ML PtTe/1 ML PtTe$_2$ films, bilayer PtTe$_2$ films were grown and then subsequently annealed at 400 °C in UHV for 1 min. The longer (5 min) annealing would lead to almost full conversion to bilayer PtTe. To convert 1 ML PtTe/1 ML PtTe$_2$ heterostruture back to PtTe$_2$ film, the heterostructure was annealed at 320 °C under Te flux. For thicker heterostructures labeled as (PtTe/PtTe$_2$)$_N$, the samples were obtained by annealing (PtTe$_2$)$_{2N}$ film where $N$ is an integer.

### ARPES measurement
Spin-integrated ARPES measurements were performed in the home laboratory with He lamp (21.2 eV) as the light source. The sample was measured at 80 K and in a vacuum better than $1 \times 10^{-10}$ Torr with energy and angular resolution of 20 meV and 0.1°, respectively. There is no change in ARPES spectra when moving the light spot around the sample (5 mm × 3 mm), indicating that the sample is homogeneous and that the ARPES data is representative of the sample.

Spin-ARPES measurements were performed using both a spin-resolved ARPES spectrometer equipped with a Xe plasma discharge lamp at Tohoku University with energy resolution of 30 meV and measurement temperature of 30 K[42], and the endstation of beam line 09U of Shanghai Synchrotron Radiation Facility (SSRF). For SSRF measurements, the energy resolution was set to 20 meV and the angular resolution is 0.2°, while the measurement temperature was 20 K.

### SHG measurements
The SHG measurements were performed using a Ti:sapphire oscillator with a center wavelength at 800 nm operating at 80 MHz repetition rate, which was used as the fundamental beam. The reflected SHG signal at wavelength of 400 nm is separated by a dichroic mirror and 400 nm centered shortpass filters in a back-reflection geometry, and then collected by a photomultiplier tube (PMT) photodetector. The polarization of the incident fundamental laser is varied by rotating a half waveplate before reaching the sample. The polarizer before PMT allows for the analysis of the polarization of the SHG by rotating its axis either parallel or perpendicular to the incident fundamental laser. The incident laser fluence is 5.2 µJ cm$^{-2}$.

### Electronic structure calculations
First-principles calculations were performed using the Vienna ab initio Simulation Package (VASP)[43] in the framework of density functional theory. The Perdew-Burke-Ernzerhof (PBE) type[44] generalized gradient approximation (GGA) was employed to address exchange and correlation effects, in conjunction with projector augmented wave (PAW) pseudopotentials. The kinetic energy cutoff of plane-wave basis sets is fixed at 350 eV, with the inclusion of self-consistent spin-orbit coupling effects. The DFT-D3 method[45] is utilized to account for the van der Waals interactions in layered PtTe$_2$. A $12 \times 12 \times 1$ $k$-point mesh was used for multi-layer thin film calculations. Lattice constants and atomic positions were fully relaxed with a force criteria of 0.02 eV/Å. Phonon dispersions and Raman spectra were computed using the frozen phonon method implemented in the Phonopy package[46], with a $3 \times 3 \times 1$ supercell and a $5 \times 5 \times 1$ uniform $k$-point mesh.

### STEM measurements
The STEM specimen was prepared using the focused ion beam (FIB) instrument. Multilayer PtTe/PtTe$_2$ heterostructure was obtained by annealing (PtTe$_2$)$_5$ on the substrate. The substrate was thinned down using an accelerating voltage of 30 kV with a decreasing current from 240 pA to 50 pA, and then with a fine polishing process using an accelerating voltage of 5 kV and a current of 20 pA. Note that the few layers on the surface were etched during the thinning process, making the total number of layers less than 10. Nevertheless, the conversion of PtTe$_2$ layer into PtTe layer is still observable. The HAADF-STEM image was acquired with an FEI Titan Cubed Themis 60–300 (operated at 300 kV) with 25 mrad convergence angle and 50 pA probe current. The HAADF detector's collection angles was 48–200 mrad. Each image was acquired with 2048 × 2048 pixels and 2 µs dwell time.

## Data availability
All data needed to evaluate the conclusions in the paper are available within the article and its Supplementary Information files. All data generated during the current study are available from the corresponding author upon request.

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

## Acknowledgements

This work was supported by National Natural Science Foundation of China (Grant Nos. 52388201, 12234011, 92250305, 52025024 and 11725418), the Ministry of Science and Technology of China (Grant Nos. 2021YFA1400100, 2023YFA1406400 and 2020YFA0308800), and the New Cornerstone Science Foundation through the XPLORER PRIZE. Changhua Bao acknowledges support from the Project funded by China Science Foundation (Grant No. BX20230187) and the Shuimu Tsinghua Scholar Program. Ken Yaegashi, Katsuaki Sugawara and Takafumi Sato acknowledge support from JST-CREST (Grant No. JPMJCR18T1). The spin-ARPES measurement was carried out with the support of Shanghai Synchrotron Radiation Facility, BL09U.

## Author contributions

S.Z. designed the research project. R.F. grew the samples. R.F., Q.L., W.C., H.Z., K.Y., K.S., T.S., and S.Z. performed the ARPES measurements and analyzed the ARPES data. Y.Z. and P.Y. performed the STEM measurements. C.B. and X.T. performed the SHG measurements. J.L. and W.D. performed the numerical calculations. R.F., P.Y., and S.Z. wrote the manuscript, and all authors commented on the manuscript.

## Competing interests

The authors declare no competing interests.
