## [Transparent Peer Review file · Nature Communications]

Giant Rashba splitting in PtTe/PtTe₂ heterostructure

Corresponding Author: Professor Shuyun Zhou

Version 0:

Reviewer comments:

Reviewer #1

(Remarks to the Author)

Feng et al, report a large Rashba splitting in a PtTe/PtTe₂ heterostructure through a combined experiment study using ARPES, SHG, Raman, STEM, and supporting DFT calculations. The PtTe/PtTe₂ heterostructures are created by annealing PtTe₂ films. Their main finding is a prominent Rashba-split band ~1eV below the Fermi level with a particularly large Rashba parameter compared with other atomically thin materials. They claim the state is exclusive to the heterostructure and is explained as the result of inversion symmetry breaking. This work is thorough and well presented. However, I have several major concerns with the manuscript (both technical and regarding novelty) that need to be addressed, before the paper can be considered for publication in Nature Communications.

1) I am not entirely convinced this is a Rashba state, and the observed state doesn't originate from two distinct bands with a small band gap. This impacts the entire premise of the paper and some additional characterisation of the origin and nature of the Rashba state itself must be performed in order to discount this possibility. While it is likely as the Authors describe, the data at present is not entirely unambiguous that this is a Rashba state in the traditional sense despite the band dispersion and spin texture matching expectations for a Rashba-split state.

I list below some specific observations from the manuscript that are potentially in support of a two-state origin for the observed Rashba-split band:

- i. There are several states in PtTe₂, PtTe and PtTe/PtTe₂ that have M-shapes or Λ -shapes that together would approximate a Rashba splitting. For example, the M-shaped state at -1.5eV in Fig. 3a in PtTe₂ looks like the upper branch of a Rashba split state. The state that develops into a topological surface state with increasing thickness (-2.5eV in 2ML PtTe₂, e.g. Extended Data Fig. 10(a)) looks like a Rashba-split state but with a gap.
- ii. The differing photon energy-dependent matrix elements of the top and bottom branches (Extended Data Fig. 7) also supports a two-state origin.
- iii. In some Figures, the Rashba state of interest does appear to be gapped. For example in Fig. 4a. Whereas, in Fig. 5 it is gapless.
- iv. The Rashba state of interest looks to be present in 2ML PtTe, but with reduced spectral weight (Extended Data Fig. 10a left panel)
- v. In Figure 3, a direct comparison is drawn to a degenerate state in PtTe₂, but as there are many more bands in the PtTe/PtTe₂ scenario, can this really be interpreted as the same state but without ISB?
- vi. In the discussion of Extended Data Fig. 11, it is stated that the Rashba splitting decreases with thickness, while it is stated in the caption that the Rashba splitting only appears for N=1. The latter is more accurate. Should one not expect a similar Rashba split state for thicker films but with reduced Rashba parameters? Instead, the thicker cases look consistent with two states that have moved in energy relative to each other with increasing thickness.

To remove this ambiguity the authors need to:

- Include a more detailed discussion of the Kramers degeneracy (with e.g. a zoom in on the DFT) to clarify that it is really gapless, and perhaps include e.g. orbital character-projected calculations to better understand the crossing.
- Provide DFT calculations at different thicknesses for both PtTe₂ and PtTe i.e. 2ML PtTe to understand whether the Rashba state exists there, given it is present in Fig S10a.
- Perform detailed photon-energy dependent ARPES to understand whether this is really a Rashba or two-band state.

Finally, the Authors should also discuss exactly why the splitting is so pronounced in this system when "it is challenging to

induce a large Rashba effect in atomically thin films". Why is this different to e.g. states in heterobilayer 2H-TMDs? This is presumably due to the local out-of-plane dipole in the Te layers, but it is not explicitly stated. Understanding this would add confidence to the findings.

2) The inversion symmetry breaking (ISB), crucial for the assignment of the state of interest as a global Rashba splitting, is predominantly inferred from the results of SHG measurements. The results are compared to bilayer PtTe₂, where a negligible signal is found, in line with its centrosymmetric crystal structure. However, the presence of an SHG signal is not always a guarantee of ISB, nor is the comparison with bilayer PtTe₂ robust.

From my understanding, the lack of a centrosymmetric contribution to SHG signal is valid only under the electric-dipole approximation, which is not necessarily fully valid for non-polar crystals like the 1T-TMDs. Indeed, a recent study (arXiv: 2308:09053) of bulk PdTe₂ (also a type-II Dirac semimetal with a very similar crystal and electronic structure to PtTe₂) demonstrated pronounced 6-fold symmetric SHG signals. These were found to originate mostly from the surface where inversion symmetry is broken by the potential step. This does not directly contradict the Authors findings but needs considering.

Furthermore, as the Authors are aware, the topologically non-trivial surface electronic structure of bulk group-X TMDs is fairly well approximated by their corresponding bilayers [Deng et al, Science Bulletin 64, 15 (2019), Lin et al., PRL 124, 036402 (2020), Hlevyack et al., npj 2D Materials and Applications 5, 40 (2021)], and thus a complete suppression of SHG signal in bilayer PtTe₂ seems surprising given the prominence of the signals in the bulk counterparts.

Could the Authors comment on this and reconcile these studies? Despite these concerns, it is of my opinion that the STEM and DFT results sufficiently confirm the PtTe/PtTe₂ structure without the SHG data, and therefore inversion symmetry is undoubtably absent.

3) Novelty. This manuscript is motivated largely by spintronic applications. However, this is a metallic system where the state is buried 1 eV below EF, and thus has little functional application beyond the increased air stability compared to PtTe₂. A short discussion/comment of the applicability of these findings to systems more broadly would greatly strengthen this manuscript. For example, could one expect to find similar Rashba-split states elsewhere in the electronic structure that can be more easily accessed (e.g. above EF)? Can these same ISB-breaking transitions with UHV annealing be engineered in other 1T-TMD bilayers (e.g. the semiconductors?), and if so, can one expect a similarly pronounced Rashba splitting? Can the Rashba states considered here be tuned to the Fermi level in some way with alloying or strain? One could also comment on if this same state could appear in twisted bilayer PtTe₂ where the ISB can be tuned.

Minor comments

1) More details on the annealing procedure should be added. I only found one line about this in the methods, stating the PtTe₂/PtTe bilayer was achieved by annealing at 400C for 1min. Given the authors state, precise control is required. What happens with longer annealing or hotter/cooler annealing? Does the bilayer only form at 400C? In addition, the details of converting back to PtTe₂ with further annealing under Te are not mentioned.

2) In Fig. S1 AFM height profiles should be included, as well as height histogram profile. This would help confirm the films are atomically flat.

3) STEM results. Given the stacking faults that are present in the STEM, it would be nice if the authors could present results for the PtTe₂/PtTe bilayer itself. In my opinion the stacking faults would be most prominent at the heterostructure/substrate interface.

4) References. There are numerous errors in the references. For example, on page 3, line 34 – Refs 13-15 do not all pertain to GeTe as stated. There are also duplicate references (e.g. 15 and 42). References should be checked carefully (I did not check all references).

5) The arrows in Fig. 5d are not defined in the caption.

6) Some text in Figures is often too small to read on a printed page.

7) Page 6, line 81. The sentence ending "...as demonstrated in a series of samples with different thicknesses" should contain a reference to a Figure or a citation.

8) In the extended Data Figure 9 caption, it is not stated what the overlaid calculations for panel f correspond to. Also, the calculations in panel i have discontinuities.

9) Flow may also be significantly improved if Fig. 3 and 5 were combined. For example, the extra states near the Fermi level in Fig. 3e are stated as originating from PtTe, yet this is only explicitly shown in Fig. 5. More generally, the most convincing evidence for a Rashba origin is in Figure 5. The constant energy contours in Figure 3 could be relegated to the supplement to make room. This is just a suggestion and by no means required.

Reviewer #2

(Remarks to the Author)

Reviewer #3

(Remarks to the Author)

The authors report a method of synthesizing PtTe/PtTe₂ heterostructure by annealing the MBE grown PtTe₂ film. Using ARPES, they observed a giant Rashba splitting in this heterostructure. The ARPES data are in very high quality, and the experimental data are substantial. However, the discussion on the physics behind the giant Rashba phenomena is relatively insufficient. Followings are my questions about this work.

1) The lattice structure of PtTe₂ is typical 1T phase, but the lattice structure of PtTe layer is not very clear for readers. I would suggest the authors also plot the side view of PtTe/PtTe₂ lattice along the y direction on the x-z plane in Figure 1a.

2) The PtTe can be formed after annealing the grown PtTe₂ film. I am curious that if annealing the PtTe/PtTe₂ for longer time or higher temperature, can all the PtTe₂ layers transit into PtTe layers? If yes, an ARPES spectra on a PtTe film would be very necessary in Figure 3 for understanding and analyzing the ARPES spectra on PtTe/PtTe₂ heterostructure.

3) In Figure 4a, it seems that a small gap opening at the crossing point (Gamma point) of the Rashba splitting bands. Is this gap from the interlayer coupling between PtTe/PtTe₂ and PtTe/PtTe₂ bilayers? If yes, this gap may highly depend on the film thickness, for only one PtTe/PtTe₂ bilayer, it should be no gap opening.

4) The STEM images show that the PtTe/PtTe₂ may have different stacking orders. Does the Rashba size depends on the stacking order? Is that possible to form a PtTe₂ terminated layer of PtTe/PtTe₂ heterostructure? Or grow a PtTe₂ layer on PtTe/PtTe₂ heterostructure to form a PtTe₂/PtTe heterostructure? I think that the Rashba splitting size would also depend on the terminated layer of PtTe/PtTe₂.

Reviewer #4

(Remarks to the Author)

Feng et al. used MBE to synthesize PtTe₂ films on graphene-terminated 6H-SiC(0001) substrates. Through annealing, the authors claimed to form PtTe/PtTe₂ heterostructures. The conversion from a PtTe₂ bilayer with an inversion symmetry to a PtTe/PtTe₂ bilayer without an inversion symmetry was confirmed by performing SHG measurements. Regular and spin-resolved ARPES measurements reveal the presence of Rashba spin splitting in the PtTe/PtTe₂ bilayer, which is absent in the PtTe₂ bilayer. The Rashba coefficient of the PtTe/PtTe₂ bilayer was found to be as high as $\alpha_R \sim 1.8 \text{ eV}\cdot\text{\AA}$. While the paper is well-written and presents a clear narrative, the authors claimed that the PtTe/PtTe₂ bilayer could have potential applications in spintronics. However, I have several major concerns about this work. Therefore, at this stage, I cannot recommend this manuscript for publication in Nature Communications.

1. From Figs. 1a and 5g, it is clear that monolayer PtTe consists of two Pt layers. This indicates that the Pt atom density in the PtTe/PtTe₂ bilayer is significantly higher than that in a PtTe₂ bilayer. Do the authors have stronger evidence that the amount of Pt atoms in a PtTe₂ bilayer is sufficient to form a PtTe/PtTe₂ bilayer?

This question became my primary concern when examining Extended Data Figs. 9a, 9d, and 9g. After annealing and adding more Te, the PtTe₂ bilayer transforms into a PtTe₂ trilayer. This seems scientifically unreasonable.

2. This study primarily focuses on achieving the Rashba effect through a material transformation rather than introducing a new physical phenomenon. Since the Rashba effect has been extensively studied in various materials, how could this approach advance the field beyond simply applying established physics to a different material?

3. This work reported a Rashba coefficient as high as $\alpha_R = 1.8 \text{ eV}\cdot\text{\AA}$. Note that this splitting occurs at around 1 eV below the Fermi level. Since spintronic applications typically rely on efficient manipulation of spin states at or near the Fermi level, how do the authors justify the relevance of these deeply situated splitting bands for practical spintronic devices?

Have the authors explored ways to shift the Rashba bands closer to the Fermi level, enhancing their potential utility in spintronic applications?

4. In Fig. 4 and Extended Data Fig. 7, different photon energies have been used. For Figs. 4c and 4d, the bands at different binding energies (one at -0.9 eV and the other at -1.2 eV) exhibit different spin-polarized directions. However, in Fig. 4h, the spin-polarization band map was measured at ~60 eV, making it challenging to discern the difference in spin-polarized directions between the bands at -0.9 eV and -1.2 eV. Could the authors clarify why the photon energy of ~60 eV is necessary for this measurement? Please also explain why the photon energy of ~50 eV is not used for the spin-polarization band map in Fig. 4h.

5. The authors attribute the need for different photon energies to probe the upper and the lower Rashba bands to the matrix element effect. However, this explanation seems unclear. While the matrix element effect is related to the geometry of the experimental setup and the photon energy, it primarily influences the intensity and visibility of ARPES spectra. Observing the bands of the same spin-splitting system at different photon energies is usually unnecessary. Could the authors explain the underlying physical mechanisms responsible for this phenomenon?

6. The homogeneity of the MBE-grown films/heterostructures is confirmed by performing ARPES by moving the light spot. However, due to the relatively large size of the light spot in regular ARPES, this method may not fully capture the smaller-scale inhomogeneity within the sample. The authors should use nano-ARPES to map over the film.

Version 1:

Reviewer comments:

Reviewer #1

(Remarks to the Author)

The authors have completed extensive additional measurements and theory calculations that I requested in the first round of review. These have significantly strengthened the paper, and I am happy for the paper to be published in Nature Communications.

Reviewer #2

(Remarks to the Author)

Reviewer #3

(Remarks to the Author)

The authors have added more experimental results and calculations in the revised manuscript, and also fully addressed my questions. With these new data and revisions, this manuscript is worth for publication. Therefore, I would recommend the publication of this manuscript in Nature Communications.

Reviewer #4

(Remarks to the Author)

I carefully went through the response letter and the revised manuscript. The authors have fully addressed all my comments. Therefore, I recommend this manuscript for publication in Nature Communications.

We thank all reviewers for valuing the scientific merits of our work. We also appreciate the constructive comments from all reviewers, which help us further improve our manuscript. In response to these comments/suggestions, we have performed extensive experimental measurements and theoretical calculations, through which the Rashba effect is further clarified.

Below please see our point-by-point response to the reviewers' comments.

Reviewer #1 (Remarks to the Author):

Feng *et al*, report a large Rashba splitting in a PtTe/PtTe₂ heterostructure through a combined experiment study using ARPES, SHG, Raman, STEM, and supporting DFT calculations. The PtTe/PtTe₂ heterostructures are created by annealing PtTe₂ films. Their main finding is a prominent Rashba-split band $\sim 1\text{eV}$ below the Fermi level with a particularly large Rashba parameter compared with other atomically thin materials. They claim the state is exclusive to the heterostructure and is explained as the result of inversion symmetry breaking. **This work is thorough and well presented.** However, I have several major concerns with the manuscript (both technical and regarding novelty) that need to be addressed, before the paper can be considered for publication in Nature Communications.

Reply: We thank the reviewer for appreciating the scientific merits and quality of our work. We would also like to thank the reviewer for further raising constructive suggestions to help us improve our manuscript. Below please see our point-to-point response to the questions.

1) I am not entirely convinced this is a Rashba state, and the observed state doesn't originate from two distinct bands with a small band gap. This impacts the entire premise of the paper and some additional characterisation of the origin and nature of the Rashba state itself must be performed in order to discount this possibility. While it is likely as the Authors describe, the data at present is not entirely unambiguous that this is a Rashba state in the traditional sense despite the band dispersion and spin texture matching expectations for a Rashba-split state.

Reply: The reviewer raised a question about alternative explanation in terms of two distinct bands rather than Rashba effect. Below we provide further evidences through new supporting experimental results to unambiguously confirm the Rashba effect:

(1) We note that the Rashba effect refers to the spin splitting due to the breaking of the

inversion symmetry. The inversion symmetry breaking in PtTe/PtTe₂ is revealed by the clear SHG signal (Fig. R1a) as well as the atomic stacking sequence of the heterostructure revealed by STEM image. The corresponding spin splitting is confirmed by Spin-ARPES measurements, where the spin direction is found to be consistent with Rashba spin texture obtained from the theoretical calculations (Fig. R1b-d). These experimental evidences provide strong evidences that the observed two bands are induced by the Rashba effect.

Fig. R1: (a) Rotational anisotropy SHG pattern of PtTe/PtTe₂ heterostructure. (b) ARPES dispersion image measured on PtTe/PtTe₂. (c-d) Spin-resolved EDCs. Red (blue) curves denote spin-up (spin-down) components. Lower panel: corresponding extracted spin polarization.

(2) The orbital characters of these spin-splitting bands are elaborated by the DFT calculations. The orbital-projected band structure of PtTe/PtTe₂ heterostructure shows a significant orbital hybridization between PtTe₂ and PtTe (Fig. R2a), and the calculated spin direction is always perpendicular to the in-plane momentum (Fig. R2b-e), in good agreement with our Spin-ARPES experimental results. This rules out the possibility of two distinct bands with a small band gap.

Fig. R2: (a) Orbital-projected band structure of PtTe/PtTe₂ heterostructure. (b) Schematic illustration for Rashba band. (c-d) The calculated spin texture at the top branches of Rashba band (c), near crossing (d), and bottom branches of Rashba band (e), as marked in (b).

(3) We note that if these bands have trivial origin, the charge doping would likely lead to a rigid shift of the Fermi level, resulting in a constant amplitude of the band splitting. While if these bands are induced the Rashba effect associated with the structural polarity (inversion symmetry breaking), the electron doping would dramatically modify the polarity, leading to distinct evolution of the band splitting. To soundly address these issues, we performed surface electron doping *via K* deposition, and then performed extensive and systematic ARPES measurements. Our experimental results in Fig. R3 show that by using surface electron doping the Rashba coefficient increases by $\sim 42\%$, which suggests that the inversion symmetry breaking effect (structural polarity) is further enhanced by surface doping induced electric field. Therefore, these results ***unambiguously confirm*** that the band splitting originates from the Rashba effect.

Fig. R3: Enhanced Rashba splitting upon surface electron doping. (a-e) Evolution of dispersion images upon surface electron doping via K deposition. The arrows mark the energy separation of the spin splitting at that momentum. (f) Energy distribution curves measured at the same momentum as indicated in (a). (g) Splitting energy evolves with doping obtained at $k = 0.1 \text{ \AA}^{-1}$ from (a-e) (h) Extracted Rashba coefficients (α_R) at different doping times. (i) Schematic illustration of enhanced Rashba splitting upon surface electron doping via K deposition.

In response to the reviewer’s question, we have added these new doping-dependent ARPES data to Supplementary Fig. 11 in the revised manuscript together with related discussions, see page 15, line 167, *“The doping dependent experiment further verifies this hypothesis. Upon K surface deposition, the charge injection from the top surface enhances the charge transfer and promotes the Rashba splitting, resulting in an increase of Rashba coefficient by 42% (see Supplementary Fig. 11).”*

With these new experimental results, we have unambiguously confirmed the Rashba effect as the origin of the observed band splitting effect, and we hope that the reviewer will be satisfied with the revision.

I list below some specific observations from the manuscript that are potentially in support of a two-state origin for the observed Rashba-split band:

i. There are several states in PtTe₂, PtTe and PtTe/PtTe₂ that have M-shapes or Λ -shapes that together would approximate a Rashba splitting. For example, the M-shaped state at -1.5eV in Fig. 3a in PtTe₂ looks like the upper branch of a Rashba split state. The state that develops into a topological surface state with increasing thickness (-2.5 eV in 2ML PtTe₂, e.g. Extended Data Fig. 10(a)) looks like a Rashba-split state but with a gap.

Reply: We thank the reviewer for suggestions to perform further analysis to exclude a two-state origin. Since the prerequisite for the Rashba state is the inversion symmetry-breaking (ISB), Rashba effect in the centrosymmetric PtTe or PtTe₂ can be safely ruled out, while the pronounced Rashba splitting in PtTe/PtTe₂ results from large interlayer coupling induced orbital hybridization between PtTe and PtTe₂ layer (Fig. R4a). To systematically investigate the influence of the orbital hybridization to the band evolution, we have performed band calculations with gradually increasing layer separation (Fig. R4b-d). The calculation results reveal that the splitting decreases with increasing separation and the consequent orbital hybridization, further confirming the band splitting is due to Rashba effect. Interesting to note that when increasing the interlayer spacing by 3.0 Å, where PtTe₂ and PtTe layers have negligible hybridization, the M-shaped and Λ -shaped dispersions (as the reviewer pointed out) from PtTe and PtTe₂ layers are clearly identified while with a large energy separation and totally vanished hybridization, which is clearly different from our experimental results.

Fig. R4: (a) Real-space charge density in the Rashba state. (b-d) Electronic band structure of PtTe/PtTe₂ heterostructure obtained by adding a vacuum spacing of 0.1, 1.0 and 3.0 Å to the interlayer spacing. Different colors and circle sizes distinguish contributions from PtTe (red) and PtTe₂ (blue).

Following the suggestion of the reviewer, we have shown in Fig.R5b the zoom-in calculated band structure of PtTe/PtTe₂, and result clearly reveals that the Rashba band crossing is gapless (Fig. R5b), which again excludes the two-state origin for splitting

of band.

Fig. R5: (a) Calculated band structure of PtTe/PtTe₂ heterostructure. (b) Zoom-in band structure to determine the crossing is gapless.

ii. The differing photon energy-dependent matrix elements of the top and bottom branches (Extended Data Fig. 7) also supports a two-state origin.

Reply: We thank the reviewer for pointing out this. To clarify this issue, we have also performed orbital-projected band structure calculations, which shows that the Rashba states at the Γ point originate from the mixing of PtTe and PtTe₂ layer (Fig. R6a), and the top and bottom branches have different contributions from PtTe and PtTe₂ respectively. Since these different orbital contributions will have different matrix elements, it is quite reasonable that top and bottom branches have differing photon energy-dependent matrix elements. We would also like to point out that different photon energy-dependent matrix elements of the top and bottom branches of Rashba band are also observed in the Bi/Ag(111) alloy, where Rashba splitting is well-established (PRL 98, 186807 (2007); EPL 87, 37003 (2009)).

Fig. R6: (a) Orbital-projected band structure of PtTe/PtTe₂ heterostructure. Different colors distinguish contributions from PtTe and PtTe₂. (b-d) ARPES dispersion image of PtTe/PtTe₂ heterostructure using different photon energy of (a) 7 eV, (b) 21.2 eV, (c) 60 eV.

In response to the reviewer’s question, we have added related discussions about the intensity of the top and bottom to clarify this, please see page 12, line 138, “*We note that the matrix elements of the upper and lower bands are different because they have different contributions from PtTe and PtTe₂ layers.*”

iii. In some Figures, the Rashba state of interest does appear to be gapped. For example in Fig. 4a. Whereas, in Fig. 5 it is gapless

Reply: We thank the reviewer for this critical comment. We realize that the cut shown in Fig.4a of the previous version was slightly off the Γ point (center of Brillouin zone). In the revised manuscript, we have fixed this problem by cutting through the Γ point (see panel d in Fig. R7 below), which shows consistent gapless Rashba bands protected by time reversal symmetry.

Fig. R7: (a) ARPES map at -0.96 eV near the cross point of Rashba band. (b-f) ARPES cut along different momentum as marked in (a). Only at Γ point (cut 3) the Rashba band crossing is visible.

iv. The Rashba state of interest looks to be present in 2ML PtTe, but with reduced spectral weight (Extended Data Fig. 10a left panel)

Reply: The data for 2 ML PtTe shown in Supplementary Fig.10a (Supplementary

Fig. 13a in revised version) were obtained by annealing a 2 ML PtTe₂ sample for longer time. We note that while the majority of the sample is converted into 2 ML PtTe, it is possible to have a very small percentage of PtTe/PtTe₂. Note that the residual PtTe/PtTe₂ signal is barely detectable (Fig. R8b) and does not affect our conclusion, and the experimental data are in overall good agreement with the calculated band structure, which does not show a Rashba splitting (see Fig. R8).

Fig. R8: (a) ARPES dispersion image of 2 ML PtTe films (Data in Supplementary Figure 10a in the previous version). (b) EDC to show that the signal contribution from PtTe/PtTe₂ (pointed by gray arrow) is small enough, and the signal is dominated by 2 ML PtTe. (c) Calculated band structure of 2 ML PtTe.

v. In Figure 3, a direct comparison is drawn to a degenerate state in PtTe₂, but as there are many more bands in the PtTe/PtTe₂ scenario, can this really be interpreted as the same state but without ISB?

Reply: The colored curves are simply guides for the eyes to show the dramatic modification of the band structure after annealing. In response to the reviewer's question, we have revised this figure by adding the PtTe data as shown in Fig. R9.

Fig. R9: Comparison of the electronic structure of bilayer PtTe_2 film, $\text{PtTe}/\text{PtTe}_2$ heterostructure and bilayer PtTe film. (a), ARPES dispersion image of bilayer PtTe_2 measured along the $M-\Gamma-K$ direction. (b) Constant energy maps at Fermi level of bilayer PtTe_2 . (c-d) The same as (a,b) but for $\text{PtTe}/\text{PtTe}_2$ heterostructure. (e-f) The same as (a,b) but for bilayer PtTe .

vi. In the discussion of Extended Data Fig. 11, it is stated that the Rashba splitting decreases with thickness, while it is stated in the caption that the Rashba splitting only appears for $N=1$. The latter is more accurate. Should one not expect a similar Rashba split state for thicker films but with reduced Rashba parameters? Instead, the thicker cases look consistent with two states that have moved in energy relative to each other with increasing thickness.

Reply: Regarding the thickness dependence, we would like to point out that for $N>1$, the effect of the broken inversion symmetry becomes strongly reduced, which is supported by the SHG signal shown in Fig. 2(e). This would consequently lead to a strongly reduced Rashba effect. Note that even when there is no Rashba effect, the M-shaped and Λ -shaped dispersions (as the reviewer pointed out) from PtTe and PtTe_2 with a large energy separation can still be observed. However, this is totally different from the Rashba effect observed in $N = 1$ case.

In response to the reviewer's comment, we have revised the discussion of Supplementary Fig. 4 (Supplementary Fig. 11 in previous version), please see page 16,

lines 179, “ARPES measurements on multilayer heterostructures, which are obtained by annealing thicker PtTe₂ films, show a negligible Rashba splitting with thickness increasing (see Supplementary Fig.4).”

To remove this ambiguity the authors need to:

-Include a more detailed discussion of the Kramers degeneracy (with e.g. a zoom in on the DFT) to clarify that it is really gapless, and perhaps include e.g. orbital character-projected calculations to better understand the crossing.

Reply: We thank the reviewer for the kind suggestion to improve our manuscript. Following the reviewer’s suggestion, we have performed detailed DFT calculations near the crossing point to confirm that it is gapless (pointed by red arrow in Fig. R10b). The orbital-projected band structure (Fig. R11a) reveals that the Rashba states near the Γ point predominantly originate from the PtTe₂ layer. Along the high-symmetry directions, the atomic contribution from PtTe gradually increases, indicating a significant orbital hybridization between PtTe₂ and PtTe, while such hybridization is absent for the case without Rashba effect. Moreover, the calculated spin polarization of splitting band is always perpendicular to the in-plane momentum (Fig. R11b-e), which is consistent with the Rashba texture. In summary, the calculations provide convincing evidences for the Rashba effect in PtTe/PtTe₂ heterostructure.

Fig. R10: (a) Calculated band structure of PtTe/PtTe₂ heterostructure. (b) Zoom-in band structure to determine the crossing is gapless.

Fig. R11: (a) Orbital-projected band structure of PtTe/PtTe₂ heterostructure. (b) Schematic illustration for Rashba band. (c-d) The calculated spin texture at the top branches of Rashba band (c), near crossing (d) and bottom branches of Rashba band (e) as marked in (b).

-Provide DFT calculations at different thicknesses for both PtTe₂ and PtTe i.e. 2ML PtTe to understand whether the Rashba state exists there, given it is present in Fig S10a.

Reply: Following the reviewer's suggestion, we show in Fig. R12 the comparison of calculated electronic structures for 2 ML PtTe₂, 2 ML PtTe and 1 ML PtTe/1 ML PtTe₂ heterostructure. It is clear that there is no Rashba splitting in 2 ML PtTe.

Fig. R12: (a) Calculated band structure of 2 ML PtTe₂. (b) Calculated band structure of 2 ML PtTe. (c) Calculated band structure of 1 ML PtTe/1 ML PtTe₂.

-Perform detailed photon-energy dependent ARPES to understand whether this is really

a Rashba or two-band state.

Reply: Following the reviewer’s suggestion, we have measured the photon-energy dependent ARPES and carefully compared different sets of data cutting through the Γ point. It is clear that the gapless feature of the band at the Γ point and the obtained band splitting is robust at different probing photon energies (Fig. R13).

Fig. R13: ARPES dispersion image of PtTe/PtTe₂ heterostructure using different photon energy of (a) 7 eV, (b) 21.2 eV, (c) 60 eV.

Finally, the Authors should also discuss exactly why the splitting is so pronounced in this system when “it is challenging to induce a large Rashba effect in atomically thin films”. Why is this different to e.g. states in heterobilayer 2H-TMDs? This is presumably due to the local out-of-plane dipole in the Te layers, but it is not explicitly stated. Understanding this would add confidence to the findings.

Reply: We thank reviewer for the kind suggestions. The pronounced Rashba effect is attributed to the strong coupling (orbital hybridization) between PtTe and PtTe₂ layers (Fig. R14a). More importantly, PtTe and PtTe₂ have very similar crystal symmetry and lattice constant, allowing better alignment of Te atoms on both sides of the heterostructure, which is crucial to obtain a pronounced orbital hybridization. Below we have performed DFT calculations and doping dependent experiments to verify this interlayer coupling strength. Such similar lattice and strong coupling explain why the Rashba effect here could be stronger than other hetero-bilayer 2H-TMDs.

Fig. R14: (a) Real-space distributions charge density in the Rashba state. (b-d) Electronic band structure of PtTe/PtTe₂ heterostructure obtained by adding a vacuum spacing of 3.0, 1.0 and 0.1 Å to the interlayer spacing. Different colors and circle sizes distinguish contributions from PtTe (red) and PtTe₂ (blue).

To evaluate the interlayer coupling, we have performed theoretical calculations to reveal the real-space charge density in the Rashba state (Fig. R14a), which provides theoretical insight into the spatial distribution of the electronic charge density. The charge density on two Te atoms in the bottom PtTe₂ monolayer shows an asymmetric behavior and extends into the top PtTe layer. This is attributed to the strong Te-Te interaction across the van der Waals (vdW) interface, which has been revealed in the previous part of orbital projected band structure (Fig. R11a), allowing for charge transfer between the layers.

To investigate how the interlayer coupling influences the Rashba state, we regulate this coupling by increasing the layer separation and calculated the corresponding band structures. When the increase of the interlayer distance is beyond 3.0 Å, the orbital hybridization between PtTe and PtTe₂ completely diminishes, showing a negligible band splitting at Γ point (pointed by arrow in Fig. R14b). Conversely, when the layer distance decreases, the orbital hybridization across the vdW vacuum becomes more significant, leading to substantial enhancement of the Rashba splitting (Fig. R14c, d).

Along this path, we further proposed an electron doping study with K deposition. The charge doping strongly enhances the charge transfer thus enlarges the Rashba splitting, as revealed by our experimental results (Fig. R15). Specially, the exacted Rashba coefficient α_R increases by $\sim 42\%$ upon surface electron doping via K deposition (Fig. R15h), further supporting that charge transfer is responsible for the Rashba state.

Fig. R15: Enhanced Rashba splitting upon surface electron doping. (a-e) Evolution of dispersion images upon surface electron doping via K deposition. The arrows mark the energy separation of the spin splitting at that momentum. (f) Energy distribution curves measured at the same momentum as indicated in (a). (g) Splitting energy evolves with doping obtained at $k = 0.1 \text{ \AA}^{-1}$ from (a-e). (h) Extracted Rashba coefficient α_R at different doping time. (i) Schematic illustration of enhanced Rashba splitting upon surface electron doping via K deposition.

These observations further highlight the influence of the vdW interaction and orbital hybridization on the electronic properties of the PtTe/PtTe₂ heterostructures, particularly in relation to the Rashba states. Such understanding is crucial for designing and engineering materials with desired electronic properties and functionalities.

In response to the reviewer's comment, we have added the calculated charge distribution of PtTe/PtTe₂ heterostructure and doping experiment in Fig. 5g and see Supplementary Fig. 11 and related discussion on page15, lines 164, of the revised manuscript, *“The charge redistribution in the heterostructure resulted from strong interlayer coupling which is responsible for the Rashba states, and the charge transfer between PtTe and PtTe₂ is verified by calculated real-space charge density in the Rashba state (Fig. 5g). The doping dependent experiment further verifies this hypothesis. Upon K surface deposition, the charge injection from the top surface*

enhances the charge transfer and promotes the Rashba splitting, resulting in an increase of the Rashba coefficient by ~42% (see Supplementary Fig. 9)."

2) The inversion symmetry breaking (ISB), crucial for the assignment of the state of interest as a global Rashba splitting, is predominantly inferred from the results of SHG measurements. The results are compared to bilayer PtTe₂, where a negligible signal is found, in line with its centrosymmetric crystal structure. However, the presence of an SHG signal is not always a guarantee of ISB, nor is the comparison with bilayer PtTe₂ robust.

From my understanding, the lack of a centrosymmetric contribution to SHG signal is valid only under the electric-dipole approximation, which is not necessarily fully valid for non-polar crystals like the 1T-TMDs. Indeed, a recent study (arXiv: 2308:09053) of bulk PdTe₂ (also a type-II Dirac semimetal with a very similar crystal and electronic structure to PtTe₂) demonstrated pronounced 6-fold symmetric SHG signals. These were found to originate mostly from the surface where inversion symmetry is broken by the potential step. This does not directly contradict the Authors findings but needs considering.

Furthermore, as the Authors are aware, the topologically non-trivial surface electronic structure of bulk group-X TMDs is fairly well approximated by their corresponding bilayers [Deng *et al.*, Science Bulletin 64, 15 (2019), Lin *et al.*, PRL 124, 036402 (2020), Hlevyack *et al.*, npj 2D Materials and Applications 5, 40 (2021)], and thus a complete suppression of SHG signal in bilayer PtTe₂ seems surprising given the prominence of the signals in the bulk counterparts.

Could the Authors comment on this and reconcile these studies? Despite these concerns, it is of my opinion that the STEM and DFT results sufficiently confirm the PtTe/PtTe₂ structure without the SHG data, and therefore inversion symmetry is undoubtedly absent.

Reply: We thank the reviewer for suggestion to exclude other possible explanations and the reviewer's conclusion that inversion symmetry is really broken as "the STEM and DFT results sufficiently confirm the PtTe/PtTe₂ structure without the SHG data, and therefore inversion symmetry is undoubtedly absent." After careful analysis, we believe that the SHG signal can reveal the ISB because of the following reasons:

(1) For materials with inversion symmetry, SHG signal arising from quadrupolar type terms, which is much weaker. Since our incident laser fluence is not large ($5.2 \mu\text{J cm}^{-2}$), the SHG signal from quadrupolar type terms is very weak and may be buried by the noise, thus no clear feature could be detected as observed in bilayer PtTe₂. In our PtTe/PtTe₂ heterostructure, there is a large SHG signal with clear six-fold symmetry. Such large SHG signal could not be attributed to quadrupolar type terms but rather should be attributed to the ISB.

(2) As for the SHG signal in bulk PdTe₂, the SHG signal increases with time after cleaving, which indicates the SHG signal could possibly results from the oxidation at PdTe₂ surface with TeO₂ skin layer or surface-to-bulk diffusion of defects (arXiv: 2308:09053). However, for our PtTe/PtTe₂ heterostructure, the excellent air-stability has been confirmed by Supplementary Fig. 13, giving confidence that SHG signal is intrinsic to the heterostructure and reflects the ISB.

(3) A complete suppression of SHG signal in bilayer PtTe₂ film is reasonable. Second-order nonlinear responses in topological materials are related to Berry connection (Sci. Adv. 2, e1501524 (2016); Nat. Rev. Phys. 4, 33 (2022)), where multifold fermion is needed. As revealed by ARPES data (Fig. R16, adopted from Science Bulletin 64, 15(2019)), the Dirac point is gapped as the thickness reduces down to 2 ML. Therefore, there is no multifold fermion contributing to SHG signal in bilayer PtTe₂.

Fig. R16: Thickness dependent band structure of PtTe₂. Reprinted from Science Bulletin 64, 15(2019).

In response to reviewer's suggestion, we have revised our manuscript, see page 9, line 108, *“Nevertheless, the observation of strong SHG in the thinnest PtTe/PtTe₂ heterostructure together with STEM results confirms that the inversion symmetry is broken”* and page 16, line 176, *“the PtTe/PtTe₂ heterostructure shows excellent stability with robust Rashba splitting under exposure to the air for days (see Supplementary Fig.*

13), which is critical for device application and indicates our ex-situ SHG measurement detects the intrinsic signal without surface oxidation as in PdTe₂ case (arxiv2328.09063).”

3) Novelty. This manuscript is motivated largely by spintronic applications. However, this is a metallic system where the state is buried 1 eV below E_F , and thus has little functional application beyond the increased air stability compared to PtTe₂. A short discussion/comment of the applicability of these findings to systems more broadly would greatly strengthen this manuscript. For example, could one expect to find similar Rashba-split states elsewhere in the electronic structure that can be more easily accessed (e.g. above E_F)? Can these same ISB-breaking transitions with UHV annealing be engineered in other 1T-TMD bilayers (e.g. the semiconductors?), and if so, can one expect a similarly pronounced Rashba splitting? Can the Rashba states considered here be tuned to the Fermi level in some way with alloying or strain? One could also comment on if this same state could appear in twisted bilayer PtTe₂ where the ISB can be tuned.

Reply: We thank the reviewer for the kind suggestions. In response to the reviewer’s suggestion, we have added discussions of potential applicability on page16, lines 185 of the revised manuscript, *“These results could have potential applications in spintronics devices. The giant Rashba states in PtTe/PtTe₂ could be possibly moved towards the Fermi level by Ir doping, as previous work has demonstrated the Fermi energy can be tuned in Ir_xPt_{1-x}Te with Pt dopants while maintaining the same crystal structure and band feature (Adv. Mater. 30,1801556(2018)). Moreover, this strategy to induce a Rashba states may apply to a broad of materials that could be constructed into TMMC/TMDC heterostructure. For example, NiTe/NiTe₂ (Phys. Status Solidi B 257,1900224(2020)) or CoTe/CoTe₂ (Bull. Soc. Chim. Belg. 80,107(1971)), in which the partner compounds share similar symmetry and lattice constant, have potential to obtain polar state towards Rashba effect along this pathway.”*

Minor comments

1) More details on the annealing procedure should be added. I only found one line about this in the methods, stating the PtTe₂/PtTe bilayer was achieved by annealing at 400°C for 1 min. Given the authors state, precise control is required. What happens with longer annealing or hotter/cooler annealing? Does the bilayer only form at 400°C? In addition, the details of converting back to PtTe₂ with further annealing under Te are not mentioned.

Reply: As for annealing under different temperature or annealing time, the longer annealing (5 minutes) would lead to almost full conversion to bilayer PtTe, while shorter annealing would leave some residual bilayer PtTe₂. In response to the reviewer’s suggestion, we have added more details to the methods session, see page 17, line 207, *“For fabrication of 1 ML PtTe/1 ML PtTe₂ films, bilayer PtTe₂ films were grown and then subsequently annealed at 400 °C in UHV for 1 minute. The longer (5 minutes) annealing would lead to almost full conversion to bilayer PtTe. To convert 1 ML PtTe/1 ML PtTe₂ heterostructure back to PtTe₂ film, the heterostructure was annealed at 320° under Te flux.”*

2) In Fig. S1 AFM height profiles should be included, as well as height histogram profile. This would help confirm the films are atomically flat.

Reply: We thank the reviewer for the nice suggestion. To confirm that the films are atomically flat, we have further measured the surface topography using scanning tunneling microscope (STM), as shown in Fig. R17. The STM topography data show that our films are atomically flat, and the height histogram profile is consistent with 2ML PtTe₂ (Fig. R17b). In response to reviewer’s suggestion, we have revised our manuscript by adding STM image into Supplementary Fig. 2 on page 28.

Fig. R17: Surface topography of 2ML PtTe₂ film. (a) STM image of 2 ML PtTe₂ before annealing. (b) The height histogram profile along black line in (a). The step height 1.3 nm is consistent with 2 ML PtTe₂.

3) STEM results. Given the stacking faults that are present in the STEM, it would be nice if the authors could present results for the PtTe₂/PtTe bilayer itself. In my opinion

the stacking faults would be most prominent at the heterostructure/substrate interface.

Reply: We would like to clarify that although stacking faults might present in thicker heterostructure, the stacking order of PtTe/PtTe₂ is much simpler, and we believe that the PtTe is on the top while PtTe₂ on the bottom due to the following reasons. Firstly, when annealing bilayer PtTe₂, the top layer is more likely to lose Te and convert into PtTe at the very beginning. Secondly, in our stability test, PtTe/PtTe₂ heterostructure shows a notable improved stability compared with that of PtTe₂, similar as the PtTe case (Fig. R18). This suggests that the top PtTe layer could protect the bottom PtTe₂ layer. Thirdly, when depositing K atoms, the band contributed by PtTe₂ disappears (Fig. R19), indicating PtTe₂ is on the bottom that photoelectron is vulnerable to surface impurity.

As for the STEM image of PtTe/PtTe₂ bilayer itself, it is pity that the thinning process to prepare the STEM sample leads to serious sample damages for the bilayer sample, as we have noted in the method, line 247, page 19: *“Note that the few layers on the surface were etched during the thinning process, making the total number of layers less than 10”*.

Fig. R18: Stability test of PtTe₂, PtTe/PtTe₂ and PtTe samples. (a-c) Data acquired in situ after the film growth for 2 ML PtTe (a), PtTe/PtTe₂ (b) and 2 ML PtTe₂ (c). (d-f) Data acquired after air exposure for two days. The PtTe (d) and PtTe/PtTe₂ (e) samples show a notable improved stability as comparing with that of PtTe₂ (f).

Fig. R19: (a) ARPES dispersion of PtTe/PtTe₂ before K doping. The white arrow points the band mainly contributed from PtTe₂. (b) ARPES dispersion of PtTe/PtTe₂ after K doping. (c) Calculated dispersion for PtTe/PtTe₂ heterostructure. Different colors and circle sizes distinguish contributions from PtTe and PtTe₂.

4) References. There are numerous errors in the references. For example, on page 3, line 34 – Refs 13-15 do not all pertain to GeTe as stated. There are also duplicate references (e.g. 15 and 42). References should be checked carefully (I did not check all references).

Reply: We thank the reviewer for the careful reading. The mentioned references have been corrected in the revised version. In addition, we have checked other references carefully.

5) The arrows in Fig. 5d are not defined in the caption.

Reply: We thank the reviewer for pointing out. We have added the caption of Fig. 5d, see page 13, “Brown and blue arrows indicate electronic bands from PtTe and PtTe₂, respectively.”

6) Some text in Figures is often too small to read on a printed page.

Reply: We thank the reviewer's suggestion to improve our presentation. We have enlarged the text in the Figures to make them more friendly to read.

7) Page 6, line 81. The sentence ending "...as demonstrated in a series of samples with different thicknesses" should contain a reference to a Figure or a citation.

Reply: We thank the reviewer for this kind suggestion. We have added the reference to Figure in page 6, line 84, "*as demonstrated in a series of samples with different thicknesses (see Supplementary Fig. 4)*".

8) In the extended Data Figure 9 caption, it is not stated what the overlaid calculations for panel f correspond to. Also, the calculations in panel i have discontinuities.

Reply: We thank the reviewer for pointing out this. We have added the caption to stated calculation band structure, see Supplementary Fig. 12, "*d-f, The same as (a-c) but for PtTe/PtTe₂ obtained after annealing in UHV. The superposed calculated PtTe/PtTe₂ band structure is shown in (f).*".

The discontinuities in panel i are due to spectral weight added to band structure, we have replaced the calculations using the same style as panel c.

9) Flow may also be significantly improved if Fig. 3 and 5 were combined. For example, the extra states near the Fermi level in Fig. 3e are stated as originating from PtTe, yet this is only explicitly shown in Fig. 5. More generally, the most convincing evidence for a Rashba origin is in Figure 5. The constant energy contours in Figure 3 could be relegated to the supplement to make room. This is just a suggestion and by no means required.

Reply: We thank the reviewer for the suggestion. To inform reader the extra band near Fermi level originating from PtTe, we have added PtTe ARPES data for comparison in Fig. 3, as shown in Fig. R20.

Fig. R20: Comparison of the electronic structure of bilayer PtTe_2 film, $\text{PtTe}/\text{PtTe}_2$ heterostructure and bilayer PtTe film. (a), ARPES dispersion image of bilayer PtTe_2 measured along the $M-\Gamma-K$ direction. (b) Constant energy maps at Fermi level of bilayer PtTe_2 . (c-d) The same as (a,b) but for $\text{PtTe}/\text{PtTe}_2$ heterostructure. (e-f) The same as (a,b) but for bilayer PtTe .

Reviewer #2 (Remarks to the Author):

Reply: We thank the reviewer for the time and efforts to review our manuscript, and for providing constructive suggestions to help us further improve our manuscript.

Reviewer #3 (Remarks to the Author):

The authors report a method of synthesizing $\text{PtTe}/\text{PtTe}_2$ heterostructure by annealing the MBE grown PtTe_2 film. Using ARPES, they observed a giant Rashba splitting in this heterostructure. **The ARPES data are in very high quality, and the experimental data are substantial.** However, the discussion on the physics behind the giant Rashba

phenomena is relatively insufficient. Followings are my questions about this work.

Reply: We thank the reviewer for appreciating the scientific merits and quality of our work, and for raising suggestions to help us improve the manuscript. We have performed more DFT calculations and further ARPES measurements upon electron doping to reveal the physics behind the obtained giant Rashba phenomena, where a strong interlayer coupling induced charge transfer results in a giant Rashba effect.

Fig. R21: (a) Real-space charge density in the Rashba state. (b-d) Electronic band structure of PtTe/PtTe₂ heterostructure obtained by adding a vacuum spacing of 3.0, 1.0 and 0.1 Å to the interlayer spacing. Different colors and circle sizes distinguish contributions from PtTe (red) and PtTe₂ (blue).

The interlayer coupling in PtTe/PtTe₂ could be evaluated by calculating the real-space charge density in the Rashba state (Fig. R21a). Charge density on two Te atoms in the bottom PtTe₂ monolayer shows an asymmetric behavior and extends into the top PtTe layer. This can be attributed to the strong Te-Te interaction across the van der Waals (vdW) interface, which have been revealed in the previous part of orbital projected band structure (Fig. R19c), allowing for charge transfer between the layers.

In order to know how this interlayer coupling influences Rashba state, we regulate this coupling by increasing layer separation and calculated the band structure evolution. When the layer distance is artificially increased beyond 3.0 Å, the orbital hybridization between PtTe and PtTe₂ diminishes, showing a negligible band splitting at Γ point (pointed by arrow in Fig. 21b). Conversely, when the layer distance decreases, the orbital hybridization across the vdW vacuum becomes significant, leading to substantial Rashba splitting (Fig. 21c,d).

Along this path, we further propose that electron doping would enhance the charge transfer thus enlarge the Rashba splitting, which has been proved by our doping

dependent experimental results (Fig. R22). Through alkali metal (K atom) doping, the splitting of Rashba band becomes larger (Fig. R22f). The exacted Rashba coefficient α_R also increase $\sim 42\%$ upon surface electron doping via K deposition (Fig. R22h), further supporting that charge transfer is responsible for the Rashba state.

Fig. R22: Enhanced Rashba splitting upon surface electron doping. (a-e) Evolution of dispersion images upon surface electron doping via K deposition. The arrows mark the energy separation of the spin splitting at that momentum. (f) Energy distribution curves measured at the same momentum as indicated in (a). (g) Splitting energy evolves with doping obtained at $k = 0.1 \text{ \AA}^{-1}$ from (a-e). (h) Extracted Rashba coefficient α_R at different doping time. (i) Schematic illustration of enhanced Rashba splitting upon surface electron doping via K deposition.

These observations highlight the influence of the vdW interaction and orbital hybridization on the electronic properties of the PtTe/PtTe₂ heterostructures, particularly in relation to the Rashba states. Such understanding is crucial for designing and engineering materials with desired electronic properties and functionalities.

In response to the reviewer's comment, we have added the calculated charge distribution of PtTe/PtTe₂ heterostructure and doping experiment in Fig. 5g and Supplementary Fig. 11 of the revised manuscript and related discussion on page 15, lines 164 of the revised manuscript, *"The charge redistribution in the heterostructure*

resulted from strong interlayer coupling which is responsible for the Rashba states, and the charge transfer between PtTe and PtTe₂ is verified by calculated real-space charge density in the Rashba state (Fig. 5g). The doping dependent experiment further verifies this hypothesis. Upon K surface deposition, the charge injection from the top surface enhances the charge transfer and promotes the Rashba splitting, resulting in an increase of Rashba coefficient by ~42% (see Supplementary Fig. 9).”

1) The lattice structure of PtTe₂ is typical 1T phase, but the lattice structure of PtTe layer is not very clear for readers. I would suggest the authors also plot the side view of PtTe/PtTe₂ lattice along the y direction on the x-z plane in Figure 1a.

Reply: We thank the reviewer for this kind suggestion. To give a detail lattice structure of PtTe, we have provided y- and z-directional atomic structures (Fig. R23) in Supplementary Fig. 1 in page 27.

Fig. R23: (a) Top view of PtTe along z direction. (b) Side view of PtTe along y direction. (c) Side view of PtTe along x direction. (d) 3D view of PtTe. (e-h) The same as (a-d) but for PtTe₂.

2) The PtTe can be formed after annealing the grown PtTe₂ film. I am curious that if annealing the PtTe/PtTe₂ for longer time or higher temperature, can all the PtTe₂ layers transit into PtTe layers? If yes, an ARPES spectra on a PtTe film would be very necessary in Figure 3 for understanding and analyzing the ARPES spectra on PtTe/PtTe₂ heterostructure.

Reply: We thank the reviewer for the nice suggestion. Motivated by the comments from the reviewer, we have performed systematic annealing experiment. By annealing the PtTe₂ for longer time, we acquired bilayer PtTe (note: 1 minute annealing for PtTe/PtTe₂,

while 5 minutes annealing for bilayer PtTe). In respond to reviewer's suggestion, the band structure of PtTe has been added to Fig. 3 to better elaborate the origin of band in PtTe/PtTe₂ (Fig. R24)

Fig. R24: Comparison of the electronic structure of bilayer PtTe₂ film, PtTe/PtTe₂ heterostructure and bilayer PtTe film. (a), ARPES dispersion image of bilayer PtTe₂ measured along the M-Γ-K direction. (b) Constant energy maps at Fermi level of bilayer PtTe₂. (c-d) The same as (a,b) but for PtTe/PtTe₂ heterostructure. (e-f) The same as (a,b) but for bilayer PtTe.

3) In Figure 4a, it seems that a small gap opening at the crossing point (Gamma point) of the Rashba splitting bands. Is this gap from the interlayer coupling between PtTe/PtTe₂ and PtTe/PtTe₂ bilayers? If yes, this gap may highly depend on the film thickness, for only one PtTe/PtTe₂ bilayer, it should be no gap opening.

Reply: We thank the reviewer for this critical comment. We realize that the cut shown in Fig. 4a of the previous version was slightly off the Γ point (center of Brillouin zone). We have fixed this problem by cutting through the Γ point (see panel d in Fig. R25 below), which shows consistent gapless Rashba band protected by time reversal symmetry.

Fig. R25: (a) ARPES map at -0.96 eV near the cross point of Rashba band. (b-f) ARPES cut along different momentum as marked in (a). Only at Γ point (cut 3) the Rashba band crossing is visible.

Moreover, to remove this ambiguity, we have further performed ARPES measurement using different photo energy (Fig. R26) and find a robust gapless Rashba band.

Fig. R26: ARPES dispersion image of PtTe/PtTe₂ heterostructure using different photon energy of (a) 7 eV, (b) 21.2 eV, (c) 60 eV.

4) The STEM images show that the PtTe/PtTe₂ may have different stacking orders. Does the Rashba size depends on the stacking order? Is that possible to form a PtTe₂ terminated layer of PtTe/PtTe₂ heterostructure? Or grow a PtTe₂ layer on PtTe/PtTe₂ heterostructure to form a PtTe₂/PtTe heterostructure? I think that the Rashba splitting size would also depend on the terminated layer of PtTe/PtTe₂.

Reply: We thank the reviewer for the nice suggestion. We note that electronic structure would not change by rigidly flipping the PtTe/PtTe₂ heterostructure, but the stacking

order may regulate the band structure. We carried out systematic studies on PtTe/PtTe₂ heterostructures with distinct stacking orders, such as AB, AC, A-B and A-C stacking orders, by performing the DFT calculations. These stacking orders possess the similar lattice parameters to the energetically favored AB stacking due to the van der Waals (vdW) nature. However, the presence of strong Te-Te interactions across the vdW interface leads to a relatively high stacking fault energy (Table R1) compared to other 2D materials like graphene and MoSe₂. The electronic band structures of these heterostructures are significantly influenced by the distinct stacking orders. Despite this, the Rashba states, which are spin-split states arising from spin-orbit coupling, are still present in the valence bands (pointed by red arrows in Fig. R27d-f). These findings indicate that the stacking order in PtTe/PtTe₂ heterostructures plays a crucial role in determining their electronic band structures and can potentially be used to engineer and control their electronic properties.

As for growing one more PtTe₂ layer on PtTe/PtTe₂ heterostructure, there is inversion center in such PtTe₂/PtTe/PtTe₂ structure (Fig. R27g), thus Rashba splitting would not exist.

Fig. R27: (a-b) Crystal structure of PtTe/PtTe₂ heterostructures with different stacking orders from side view. (d-f) Calculated band of PtTe/PtTe₂ heterostructures with different stacking orders (AC, A-B, A-C). (g) Crystal structure of one more PtTe₂ on PtTe/PtTe₂ heterostructures, where the red circle is the inversion center.

Stack order	AB	AC	A-B	A-C
Energy (meV)	0	232	146	151

Table R1. Relative energy of different stacking orders in PtTe/PtTe₂ heterostructures with reference to the AB stacking order.

Reviewer #4 (Remarks to the Author):

Feng *et al.* used MBE to synthesize PtTe₂ films on graphene-terminated 6H-SiC(0001) substrates. Through annealing, the authors claimed to form PtTe/PtTe₂ heterostructures. The conversion from a PtTe₂ bilayer with an inversion symmetry to a PtTe/PtTe₂ bilayer without an inversion symmetry was confirmed by performing SHG measurements. Regular and spin-resolved ARPES measurements reveal the presence of Rashba spin splitting in the PtTe/PtTe₂ bilayer, which is absent in the PtTe₂ bilayer. The Rashba coefficient of the PtTe/PtTe₂ bilayer was found to be as high as $\alpha_R \sim 1.8 \text{ eV} \cdot \text{\AA}$. While **the paper is well-written and presents a clear narrative**, the authors claimed that the PtTe/PtTe₂ bilayer could have potential applications in spintronics. However, I have several major concerns about this work. Therefore, at this stage, I cannot recommend this manuscript for publication in Nature Communications.

Reply: We thank the reviewer for appreciating the scientific merits and quality of our work, and for raising suggestions to help us improve the manuscript. In response to the reviewer's suggestions, we have revised our manuscript, which has led to major improvement. We hope that with these revisions, the reviewer will now be satisfied.

1. From Figs. 1a and 5g, it is clear that monolayer PtTe consists of two Pt layers. This indicates that the Pt atom density in the PtTe/PtTe₂ bilayer is significantly higher than that in a PtTe₂ bilayer. Do the authors have stronger evidence that the amount of Pt atoms in a PtTe₂ bilayer is sufficient to form a PtTe/PtTe₂ bilayer?

This question became my primary concern when examining Extended Data Figs. 9a, 9d, and 9g. After annealing and adding more Te, the PtTe₂ bilayer transforms into a PtTe₂ trilayer. This seems scientifically unreasonable.

Reply: We thank the reviewer for pointing out this. We note that during the annealing, the total atomic number of Pt is conserved. Our analysis reveals that after annealing PtTe₂ to PtTe/PtTe₂ (Fig. R28), the fraction of film coverage decreases from 86% in 2 ML PtTe₂ to 57% (calculated by the area of these films from data shown in Fig.R28), which indicates that the Pt atom from those bare areas supplement the required Pt atom

during conversion. While, the formation of 3 ML PtTe₂ after Te supply is consistent with 3 layers Pt atoms in PtTe/PtTe₂ heterostructure (2 layers of PtTe and 1 layer of PtTe₂). In response to the reviewer’s question, we have added a related discussion about the film coverage, please see page 6, line 81, “*Since PtTe consists of two Pt layers while PtTe₂ consists of only one layer, the film coverage would decrease as a result of Pt atom conservation (see Supplementary Fig. 2d,e)*”

Fig. R28: Surface topography of film. (a) STM image of bilayer PtTe₂ before annealing. The inset shows height histogram profile along red line. (b) STM image of PtTe/PtTe₂ heterostructure after annealing. The decrease of film coverage is visible.

2. This study primarily focuses on achieving the Rashba effect through a material transformation rather than introducing a new physical phenomenon. Since the Rashba effect has been extensively studied in various materials, how could this approach advance the field beyond simply applying established physics to a different material?

Reply: We thank the reviewer for the suggestion. Although there are some layered materials exhibiting Rashba effect, the Rashba effect in their monolayer counterpart would often diminish. Our work provides a new strategy to break the inversion symmetry in atomically thin films, and the inversion symmetry could be reversibly back. The ability of breaking inversion symmetry to induce a Rashba effect in two-dimensional limit is interesting. Moreover, our new doping dependent ARPES results enable us to control the Rashba coefficients upon electron doping (Fig. R29), which gives another degree of freedom to manipulate this novel band structure. We note that the PtTe/PtTe₂ heterostructure provides a precious two-dimensional model system in

which a giant Rashba spin splitting can be induced and regulated with symmetry manipulation.

Fig. R29: Enhanced Rashba splitting upon surface electron doping. (a-e) Evolution of dispersion images upon surface electron doping via K deposition. The arrows mark the energy separation of the spin splitting at that momentum. (f) Energy distribution curves measured at the same momentum as indicated in (a). (g) Splitting energy evolves with doping obtained at $k = 0.1 \text{ \AA}^{-1}$ from (a-e). (h) Extracted Rashba coefficient α_R at different doping time. (i) Schematic illustration of enhanced Rashba splitting upon surface electron doping via K deposition.

3. This work reported a Rashba coefficient as high as $\alpha_R = 1.8 \text{ eV}\cdot\text{\AA}$. Note that this splitting occurs at around 1 eV below the Fermi level. Since spintronic applications typically rely on efficient manipulation of spin states at or near the Fermi level, how do the authors justify the relevance of these deeply situated splitting bands for practical spintronic devices?

Have the authors explored ways to shift the Rashba bands closer to the Fermi level, enhancing their potential utility in spintronic applications?

Reply: We thank the reviewer for the suggestion. A possible way to tune the pronounced Rashba band toward Fermi level is by Ir doping. Since IrTe_2 and PtTe_2 share the same crystal symmetry and similar lattice constant (3.93 \AA for IrTe_2 and 4.03

Å for PtTe₂), it is possible to hole dope PtTe₂ by Ir substitution. It has been reported that Pt doping in IrTe₂ would tuning electronic band downward in energy (Adv. Mater. 30, 1801556(2018)), thus Ir doping in PtTe₂ may tune the Rashba band upward to Fermi level for spintronic applications.

In addition, we propose that constructing TMMC/TMDC heterostructure to obtain polar state may apply to other materials. There are plenty of transition metals that can form both TMMC and TMDC material. Among them, some TMMC and TMDC film has similar crystal symmetry and lattice constant. For example, NiTe₂ and NiTe (Phys. Status Solidi B 257,1900224 (2020); Nat. Commun. 14, 4945 (2023)) were both adopt hexagonal structure, and their lattice constant (3.87 Å NiTe₂ verse 3.92 Å for NiTe), allowing for alignment of Te atoms which may give rise to a strong interlayer coupling.

Moreover, we have performed the alkali metal doping experiment and find a significant enhancement of Rashba coefficient with K deposition (Fig. R29). This result indicated the large Rashba splitting could exist in other heterostructure with strong charge transferring, giving a potential application in spintronic device.

Following the suggestion, we have added discussion of potential applicability on page 16, line 185 of the revised manuscript, *“These results could have potential applications in spintronics devices. The giant Rashba states in PtTe/PtTe₂ could be possibly moved towards the Fermi level by Ir doping, as previous work has demonstrated the Fermi energy can be tuned in Ir_xPt_{1-x}Te with Pt dopants while maintaining the same crystal structure and band feature (Adv. Mater. 30,1801556(2018)). Moreover, this strategy to induce a Rashba states may apply to a broad of materials that could be constructed into TMMC/TMDC heterostructure. For example, NiTe/NiTe₂ (Phys. Status Solidi B 257,1900224(2020)) or CoTe/CoTe₂ (Bull. Soc. Chim. Belg. 80,107(1971)), in which the partner compounds share similar symmetry and lattice constant, have the potential to obtain polar state towards Rashba effect along this pathway.”*

4. In Fig. 4 and Extended Data Fig. 7, different photon energies have been used. For Figs. 4c and 4d, the bands at different binding energies (one at -0.9 eV and the other at -1.2 eV) exhibit different spin-polarized directions. However, in Fig. 4h, the spin-polarization band map was measured at ~60 eV, making it challenging to discern the difference in spin-polarized directions between the bands at -0.9 eV and -1.2 eV. Could the authors clarify why the photon energy of ~60 eV is necessary for this measurement? Please also explain why the photon energy of ~50 eV is not used for the spin-

polarization band map in Fig. 4h.

Reply: We thank the reviewer for pointing out this. We note that the spin-polarized results using 50 eV could sufficiently confirm the Rashba spin texture. The map measured using 60 eV is technical complement to reinforce our results. Most importantly, we think the existing data is consistent and supports the Rashba spin texture.

5. The authors attribute the need for different photon energies to probe the upper and the lower Rashba bands to the matrix element effect. However, this explanation seems unclear. While the matrix element effect is related to the geometry of the experimental setup and the photon energy, it primarily influences the intensity and visibility of ARPES spectra. Observing the bands of the same spin-splitting system at different photon energies is usually unnecessary. Could the authors explain the underlying physical mechanisms responsible for this phenomenon?

Reply: We thank the reviewer for pointing out this. We note that the spin analyzer needs to collect sufficiently large number of photoelectrons to obtain good signal-to-noise ratio. For the photon energy where the intensity of upper Rashba band is small, the spin polarization signal will be beyond the detection limit. Therefore, we use different photon energies to enhanced the intensity of upper or lower Rashba band to get a better signal-to-noise ratio.

6. The homogeneity of the MBE-grown films/heterostructures is confirmed by performing ARPES by moving the light spot. However, due to the relatively large size of the light spot in regular ARPES, this method may not fully capture the smaller-scale inhomogeneity within the sample. The authors should use nano-ARPES to map over the film.

Reply: We thank the reviewer for point out this. We note that if the PtTe/PtTe₂ film processes different structures (such as PtTe or PtTe₂), their corresponding band structure will be detected from the ARPES measurements. However, our ARPES data on PtTe/PtTe₂ sample is consistent with the calculated band structure, with very few additional bands (Fig. R30), which suggest that at this length scale the sample is predominately PtTe/PtTe₂. The ARPES spacial map aims to determine whether our ARPES data is representative in a large scale of the sample. In response to reviewer's suggestion, we have revised the method section regard homogeneity in page 18, line 216, *"There is no change in ARPES spectra when moving the light spot around the*

sample ($5 \text{ mm} \times 3 \text{ mm}$), indicating that the sample is homogeneous and that the ARPES data is representative of the sample.”

Fig. R30: Comparison of experiment and calculated band structure. (a) ARPES dispersion image of PtTe/PtTe₂ heterostructure along M-Γ-K direction. (b) Calculated band structure of PtTe/PtTe₂ heterostructure along M-Γ-K direction.